# PTEN negatively regulates the cell lineage progression from NG2[+] glial progenitor to oligodendrocyte via mTOR-independent signaling

Estibaliz González-Fernández[1], Hey-Kyeong Jeong[1], Masahiro Fukaya[2], Hyukmin Kim[1], Rabia R Khawaja[1], Isha N Srivastava[1], Ari Waisman[3], Young-Jin Son[1,4], Shin H Kang[1,4]*

[1]Shriners Hospitals Pediatric Research Center, Temple University Lewis Katz School of Medicine, Philadelphia, Unites States; [2]Department of Anatomy, Kitasato University School of Medicine, Sagamihara, Japan; [3]Institute for Molecular Medicine, University Medical Center of the Johannes Gutenberg University of Mainz, Mainz, Germany; [4]Department of Anatomy and Cell Biology, Temple University Lewis Katz School of Medicine, Philadelphia, United States

*For correspondence:
shin.kang@temple.edu

**Abstract** Oligodendrocytes (OLs), the myelin-forming CNS glia, are highly vulnerable to cellular stresses, and a severe myelin loss underlies numerous CNS disorders. Expedited OL regeneration may prevent further axonal damage and facilitate functional CNS repair. Although adult OL progenitors (OPCs) are the primary players for OL regeneration, targetable OPC-specific intracellular signaling mechanisms for facilitated OL regeneration remain elusive. Here, we report that OPC-targeted PTEN inactivation in the mouse, in contrast to OL-specific manipulations, markedly promotes OL differentiation and regeneration in the mature CNS. Unexpectedly, an additional deletion of mTOR did not reverse the enhanced OL development from PTEN-deficient OPCs. Instead, ablation of GSK3β, another downstream signaling molecule that is negatively regulated by PTEN-Akt, enhanced OL development. Our results suggest that PTEN persistently suppresses OL development in an mTOR-independent manner, and at least in part, via controlling GSK3β activity. OPC-targeted PTEN-GSK3β inactivation may benefit facilitated OL regeneration and myelin repair.
DOI: https://doi.org/10.7554/eLife.32021.001

## Introduction

In the mammalian central nervous system (CNS), NG2 and PDGFRα (platelet-derived growth factor receptor α)-expressing OPCs (also known as NG2[+] cells) are abundant throughout life (*Dimou and Gallo, 2015*; *Nishiyama et al., 2009*). The lineage progression from OPC to OL is a multi-step process that involves gradual changes in cell morphology and properties, from OPCs to pre-myelinating OLs (pre-OLs) to fully mature OLs, culminating in the formation of lipid-rich myelin around axons (myelinogenesis) (*Nave and Werner, 2014*). In the adult CNS, OPCs continue to serve as the primary reservoir for homeostatic maintenance of OLs by generating new myelinating OLs in the event of OL loss (*Chamberlain et al., 2016*; *Young et al., 2013*), while they proliferate and sustain their own population size (*Hughes et al., 2013*).

Although a number of intracellular signaling pathways are known to regulate OL development and myelination during early development (*Gaesser and Fyffe-Maricich, 2016*), it is not clear whether the same signaling mechanisms equally regulate continuing OL generation in the adult

CNS, as well as OL regeneration after demyelination (*Fancy et al., 2011*). Indeed, genetic ablation of proposed OL development regulators often resulted in only transient delay in early myelination (*He et al., 2017*; *Huang et al., 2013*). Moreover, in contrast to a plethora of molecules whose loss-of-function results in myelin defects, there are far fewer examples of significantly improved OL generation and regeneration in the mature CNS.

It is conceivable that there are multiple cell-stage-specific signaling mechanisms, each of which may regulate a distinct aspect of the OL development and myelination, such as OPC proliferation, a permanent exit from the OPC cell cycle, morphological change, axon recognition, and, finally, plasma membrane outgrowth and tight myelin formation. Accordingly, OPC-intrinsic signaling that regulates OPC-to-OL transition should be considered as a separate step from the OL-intrinsic mechanisms that drive myelin growth and maintenance.

The mammalian target of rapamycin (mTOR) signaling is known to be necessary for proper CNS myelination in early life (*Figlia et al., 2018*; *Lebrun-Julien et al., 2014*; *Wahl et al., 2014*; *Wood et al., 2013*), but the therapeutic potential of mTOR signaling manipulation for improved OL regeneration is not clear (*Jiang et al., 2016*; *Lebrun-Julien et al., 2014*; *McLane et al., 2017*). As an effort to identify OPC-intrinsic signaling mechanisms that promote OL generation in the mature CNS, we first determined whether the OPC-specific increase in mTOR complex 1 (mTORC1) activity enhances OL development, by OPC-specific genetic deletion of the tuberous sclerosis complex 1 (TSC1) or phosphatase and tensin homolog (PTEN), the two negative regulators of mTORC1 activity (see *Figure 1A*). Here, we report that OPC-targeted PTEN genetic ablation markedly enhanced OPC proliferation and OL differentiation, regardless of the examined CNS areas and age windows. OPC-targeted PTEN inactivation also remarkably facilitates OL regeneration and promotes remyelination after toxin-induced demyelination. However, such positive outcomes were not observed for OL-specific PTEN ablation. Moreover, in contrast to PTEN, OPC-specific TSC1 inactivation impaired OL development/survival.

The conflicting outcomes of genetic inactivations of PTEN and TSC1 suggest involvement of mTORC1-independent PTEN-downstream signaling mechanisms. Supporting this, our *Pten* and *Mtor* double cKO revealed that mTORC1 signaling is dispensable for the enhanced OL generation from the PTEN-deficient OPCs. Finally, OPC-specific ablation of glycogen synthase kinase 3β (GSK3β), another PTEN-Akt downstream target, promoted OL differentiation at a level comparable to that of PTEN-ablation.

We interpret our findings to indicate that PTEN persistently and negatively regulates the OPC-to-OL transition in an mTOR-independent manner, likely via GSK3β activity. This suggests that the inhibition of the PTEN-Akt-GSK3β pathway may efficiently promote remyelination in the adult CNS.

## Results

### OPC-targeted TSC1 inactivation is detrimental to new OL development

Akt-mTORC1 signaling regulates early OL development and myelination (*Figlia et al., 2018*; *Lebrun-Julien et al., 2014*; *Wahl et al., 2014*; *Wood et al., 2013*). Because the mTORC1 activity is under inhibitory control of the TSC1/2 complex (*Tee et al., 2002*) (See *Figure 1A*), it could be expected that *Tsc1* ablation would increase mTORC1 activities, and enhance oligodendrogenesis. However, contrary to this idea, *Tsc1* cKO with *Cnp-Cre* mice resulted in impaired myelination (*Lebrun-Julien et al., 2014*). The *Cnp-Cre* is active in mature OLs (*Lappe-Siefke et al., 2003*), but it impacts only a subset of late (i.e. GPR17-expressing) OPCs during early development (*Tognatta et al., 2017*). Therefore, we reasoned that the hypomyelination observed in *Cnp-Cre*-based *Tsc1* cKO may be attributed to long-term OL-specific defects, such as myelin maintenance failure, rather than to the OPC-to-OL transition.

To test the possibility that the OPC-specific mTORC1 activity increase promotes OL development *in vivo*, we crossed *Pdgfra-CreER*[TM] mice (tamoxifen-inducible OPC-specific Cre mice) (*Kang et al., 2010*) with *Tsc1*[f/f] and *Rosa26-EYFP (R26-EYFP)* Cre reporter mice. 4-Hydroxytamoxifen (4HT) was administered into *Pdgfra-CreER; R26-EYFP; Tsc1*[f/f] mice starting at P20 (1 mg per i.p. injection, a total of three injections for 1.5 days) and their age-matched controls (*Pdgfra-CreER; R26-EYFP*) (*Figure 1B*). This 4HT injection protocol allows EYFP-labeling of more than 75% of OPCs in both the cortex (CTX) and corpus callosum (CC) of the control mice in 4 days (P20 +4) (data not shown). Three

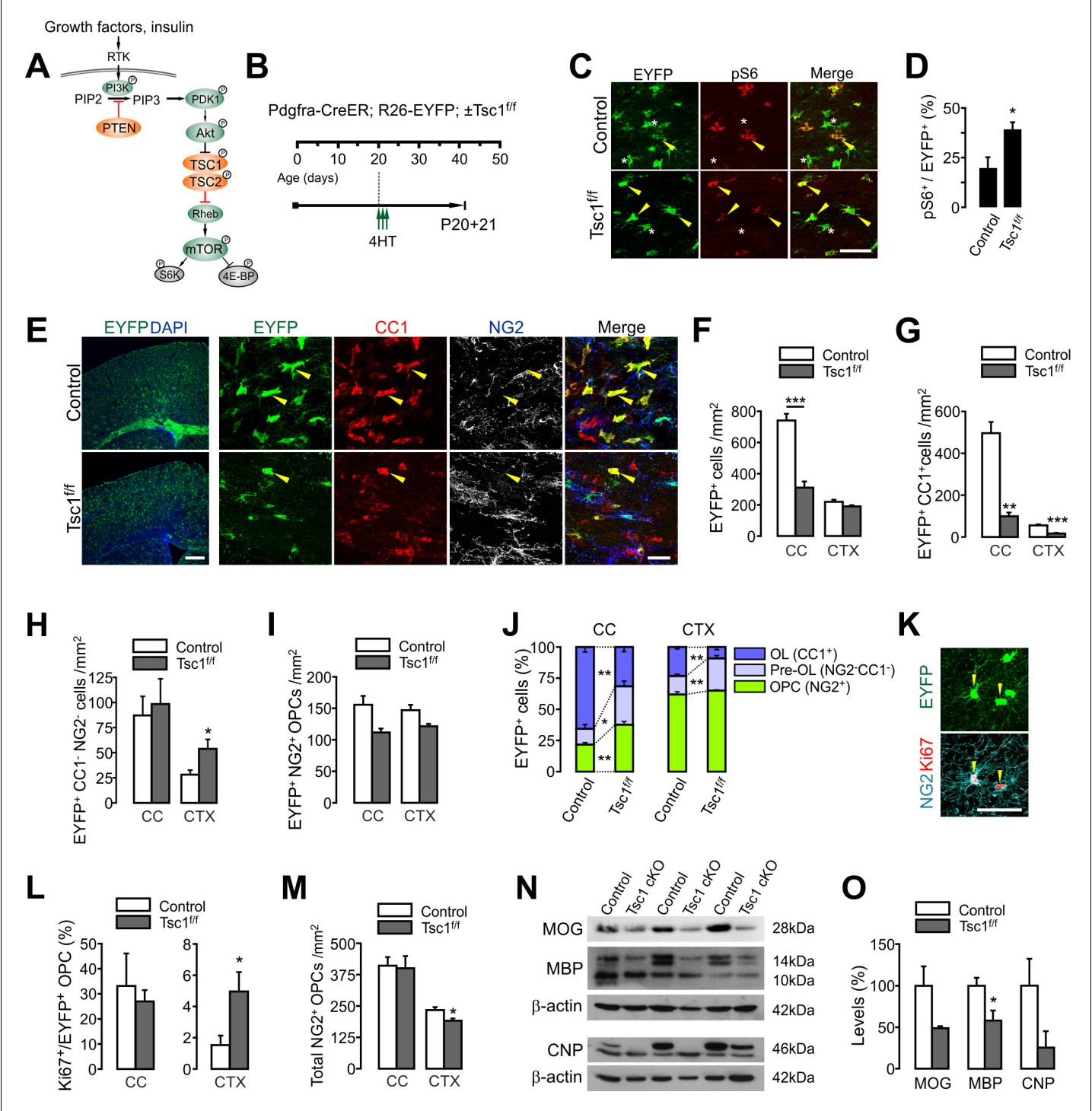

**Figure 1.** OPC-specific *Tsc1* ablation impairs oligodendrocyte development in the brain. (**A**) Schematic diagram of the Akt-mTOR signaling pathway. The TSC1/2 complex and PTEN (orange circles) negatively regulate mTOR activity, whereas other molecules in green circles positively regulate it. (**B**) Experimental scheme for 4HT administration into *Pdgfra-CreER; R26-EYFP; ±Tsc1$^{f/f}$* mice and mouse sampling. Three 4HT injections (1 mg per injection) were given between P20 and P21 (a total of 3 mg of 4HT). (**C**) Confocal images of phosphorylated S6 ribosomal protein (pS6) and EYFP$^+$ cells in the CC at P20 +21. Arrowheads and asterisks indicate EYFP$^+$ pS6$^+$ cells and EYFP$^+$ pS6$^-$ cells, respectively. Scale bar, 50 µm. (**D**) Quantification of the percentage of pS6$^+$ cells among EYFP$^+$ cells in the CC. n = 4 mice per group. (**E**) Fluorescence (left) and confocal microscopic (right) images of EYFP$^+$ cells in the control and *Tsc1* cKO mice (P20 +21). The confocal images of EYFP$^+$ cells were taken from the CTX, and show their maturation stages. Arrowheads indicate EYFP$^+$CC1$^+$ OLs. Scale bars, 500 µm (left) and 50 µm (right). (**F**) Quantification of EYFP$^+$ cells in the CC and CTX. (G - I) The numbers of EYFP$^+$CC1$^+$ OLs (**G**), EYFP$^+$CC1$^-$NG2$^-$ pre-OLs (**H**), and EYFP$^+$NG2$^+$ OPCs (**I**). (**J**) Percentages of OPC, pre-OL and OL among EYFP-labeled

*Figure 1 continued on next page*

*Figure 1 continued*

cells at P20 +21. n = 6 (control) or 3 (*Tsc1* cKO) mice for (F - J). (**K**) Cell proliferation analysis with Ki67-expressing patterns. Confocal images of EYFP⁺NG2⁺Ki67⁺ OPCs in the CTX of a *Tsc1* cKO mouse. Scale bar, 50 µm. (**L**) The percentage of Ki67⁺ cells among EYFP⁺ OPCs. n = 4 mice per group. (**M**) The number of total OPCs. (**N**) Western blot analysis of cortical lysates (P20 +21) for myelin proteins MOG, MBP, and CNP. *Tsc1^{f/f}* (control) and 4HT-administered *Pdgfra-CreER; Tsc1^{f/f}* mice (*Tsc1* cKO) were used. (**O**) Quantification of levels of myelin proteins. n = 3 mice per group for (**N, O**). Data are represented as mean ±S.E.M. *p<0.05; **p<0.01; ***p<0.001. Unpaired Student's t-test. The numerical data for the graphs are available in *Figure 1—source data 1*. Original western images are available in *Figure 1—source data 2*.
DOI: https://doi.org/10.7554/eLife.32021.002

The following source data and figure supplements are available for figure 1:

**Source data 1.** Numerical data for graphs in *Figure 1*.
DOI: https://doi.org/10.7554/eLife.32021.005
**Source data 2.** Original western blot images used for *Figure 1K*.
DOI: https://doi.org/10.7554/eLife.32021.006
**Figure supplement 1.** *Tsc1* deletion in OPCs impairs new oligodendrocyte development in the spinal cord.
DOI: https://doi.org/10.7554/eLife.32021.003
**Figure supplement 1—source data 1.** Numerical data for graphs in *Figure 1—figure supplement 1*.
DOI: https://doi.org/10.7554/eLife.32021.004

weeks after the first 4HT injection (P20 +21), there was a significant increase in the percentage of phosphorylated S6 ribosomal protein (pS6) immunopositive cells among all callosal EYFP⁺ cells in *Pdgfra-CreER; R26-EYFP; Tsc1^{f/f}* mice (p=0.042) (*Figure 1C,D*), indicating that mTORC1 activity is enhanced (*Hara et al., 1998*) as a result of effective TSC1 inactivation.

However, to our surprise, the number of EYFP⁺ cells was remarkably reduced in *Tsc1* cKO mice, particularly in the CC (*Figure 1E,F*). Co-immunostaining with antibodies against OL lineage stage markers (i.e. NG2 and CC1) (*Figure 1E*) revealed that the number of newly generated OLs (EYFP⁺-CC1⁺) was severely reduced, both in the CC (p=0.016) and CTX (p=0.0007) of the cKO mice (*Figure 1G*). In contrast, the numbers of EYFP⁺ pre-myelinating OLs (pre-OLs, NG2⁻ CC1⁻) and EYFP⁺ OPCs (NG2⁺) were not significantly affected, except for cortical pre-OLs that were increased in the *Tsc1* cKO (*Figure 1H,I*). Consequently, despite no or only a small increase in labeled OPCs and pre-OLs, the composition of EYFP-labeled cells changed to a state where the proportion of OPCs and pre-OLs became significantly larger, whereas that of mature OLs was markedly reduced in both the CC and CTX (*Figure 1J*). The altered composition of EYFP-labeled cells excludes the possibility that the marked decrease of EYFP⁺ OLs was not caused by inefficient Cre-loxP recombination in *Tsc1* cKO mice (P20 +21). The same OPC-targeted *Tsc1* cKO mice also exhibited impairment of new OL development in the spinal cord (SC) (*Figure 1—figure supplement 1*).

Our examination on the Ki67 (a marker of a proliferating cell) expression pattern in the brain (*Figure 1K*) revealed that the OPC proliferation was not impaired in these *Tsc1* cKO mice (*Figure 1L*). The percentage of Ki67⁺ cells among cortical EYFP⁺ OPCs was even higher in the cKO mice than in control (*Figure 1L*). Nonetheless, there was a slight reduction in total OPCs in the CTX of the cKO mice (*Figure 1M*). These observations indicate that the marked decrease of new OLs in the cKO mice is not due to OPC proliferation defects. Consistent with the impaired OL accumulation, the levels of MBP expression were also reduced in the CTX of *Tsc1* cKO mice (P20 +21) (*Figure 1N,O*).

These results raise two possibilities that are not mutually exclusive: Pre-OL-to-OL lineage progression is arrested, and/or newly born OLs undergo premature cell death. Although our cleaved caspase-3 staining did not reveal any increase in cell death in the cKO mice brain (data not shown), it is possible that the observation time point of cleaved caspase-3 (3 weeks after *Tsc1* deletion) is too late to detect the cell death events. The different degree of OL impairment in the CC and CTX may be related to the numbers of new OLs in the two brain areas for the examined age window (from P20 to P41). A more actively OL-generating CNS region may have been more severely affected by a premature OL loss and/or maturation arrest. Indeed, the CC is known to be more active in OL development than CTX in all examined ages in earlier studies (*Kang et al., 2010*; *Rivers et al., 2008*; *Young et al., 2013*).

Along with others' results on OL lineage cKO of *Tsc1* or *Tsc2* (*Carson et al., 2015*; *Jiang et al., 2016*; *Lebrun-Julien et al., 2014*), our results further support the idea that a complete inactivation

of TSC1 (or consequent mTORC1 hyperactivation) is detrimental to OLs, and thus would not benefit OL development or remyelination. However, the situation is not entirely clear, because another, recently published study suggests that *Tsc1*-deleted OPCs (with *Cspg4-CreER*) leads to increased myelin thickness after demyelination (*McLane et al., 2017*).

## OPC-specific PTEN inactivation promotes new OL development in both the young and mature CNS

Next, we targeted *Pten*, another upstream negative regulator of the Akt-mTORC1 signaling (See *Figure 1A*), in OPCs. *Pten* had been targeted in OL lineage cells in earlier studies, in which gene deletion was achieved using either *Olig2-Cre* (*Harrington et al., 2010*; *Maire et al., 2014*) or OL-specific Cre lines (i.e. *Cnp-Cre* and *Plp1-CreER*) (*Goebbels et al., 2010*). Those studies did not report apparent changes in OL numbers or pattern of remyelination, but abnormal myelin out-growth. However, it is possible that the cell-specific roles of PTEN in OPC were not completely revealed, because *Olig2-Cre* may affect many other neural cells besides OL lineage cells (*Maire et al., 2014*; *Masahira et al., 2006*), and OL-specific Cre mice do not impact the majority of adult OPCs (*Goebbels et al., 2010*). To investigate OPC-specific roles of PTEN, we administered 4HT into *Pdgfra-CreER; R26-EYFP; ±Pten$^{f/f}$* mice either at P20 or P45, sampling them at P41 or P75, respectively (*Figure 2A*), and measured the degree of new oligodendrogenesis from the EYFP-labeled OPCs for these two age windows (i.e. P20 ~P41 and P45 ~P75). Similar to *Tsc1* cKO, *Pten* deletion increased pS6 levels in EYFP$^+$ cells in the CC (*Figure 2B,C*), indicating that PTEN was inactivated, and as a consequence, mTORC1 activity was significantly enhanced. However, in contrast to *Tsc1* cKO, the number of EYFP$^+$ cells increased for most of the observed brain areas, including the CC and CTX (*Figure 2D,E*). The fate analysis of EYFP-labeled OPCs with co-immunostaining of cell stage markers (*Figure 2D*) revealed that there were significant increases in EYFP$^+$CC1$^+$ cells in the CTX of *Pten* cKO mice for both age-windows (p=0.0002 for P20 ~P41; p=0.0037 for P45 ~P75) (*Figure 2F*). The new OL generation was also higher in the CC of P45 +30 *Pten* cKO mice than the age-matched control (p=0.0006) (*Figure 2F*). The failure to observe a significant increase of EYFP$^+$-CC1$^+$ cells in the CC of P20 +21 *Pten* cKO (*Figure 2F*) may be due to the high rate of OL differentiation (with a larger variation) in the CC of the developing brain (*Kang et al., 2010*). Thus, facilitated OL differentiation may not have been sufficiently pronounced by the relatively short-term (i.e. 3 weeks) OPC fate-tracking. In contrast to the increased new OLs, densities of EYFP-labeled OPCs were not changed by *Pten* cKO (*Figure 2G*). Consequently, the percentage of mature OLs among EYFP-labeled OL lineage cells increased in both the CC (for P45 +30) and CTX (*Figure 2H*). These results strongly suggest that PTEN inactivation in OPCs promotes new OL development, even in the mature brain.

In order to assess changes in cell proliferation, BrdU was also given to *Pdgfra-CreER; R26-EYFP; ±Pten$^{f/f}$* mice (*Figure 2A*). We observed that there were marked increases in BrdU$^+$ cells in the *Pten* cKO brain (P20 +21) (*Figure 2I* left). Confocal imaging further confirmed that most BrdU$^+$ cells were NG2$^+$ OPCs for both age windows (*Figure 2I,J*). It was also noted that the characteristic tiled distribution of OPCs (*Hughes et al., 2013*) was disrupted by OPC-specific *Pten* cKO (*Figure 2K*). However, despite the increase of OPC proliferation, the density of total OPCs was not altered (*Figure 2L*). These observations suggest that the increased OPC proliferation directly contributes to additional OL differentiation and/or that it is a homeostatic regenerative response to the OPC reduction due to enhanced OPC-to-OL conversion in *Pten* cKO mice.

The enhanced oligodendrogenesis was also evident in both white and gray matter (WM and GM) of the ventral SC of *Pten* cKO mice, for both examined age windows (*Figure 2—figure supplement 1A–C*). In particular, after OPC-specific *Pten* ablation starting at P45, new OL accumulation was enhanced by about four-fold in the SC GM in 30 days (p=0.0001) (*Figure 2—figure supplement 1C*). However, the densities of labeled OPCs and total OPCs were not altered (*Figure 2—figure supplement 1D,E*), as they were in the brain. These results further suggest that OPC-specific PTEN inactivation effectively promotes new OL development throughout the CNS.

## OPC-targeted *Pten* ablation leads to enhanced OL accumulation

Even though OL generation is enhanced by the OPC-targeted genetic manipulation, survival of new OLs may be contingent on the availability of extracellular growth factors or other cell-extrinsic cues.

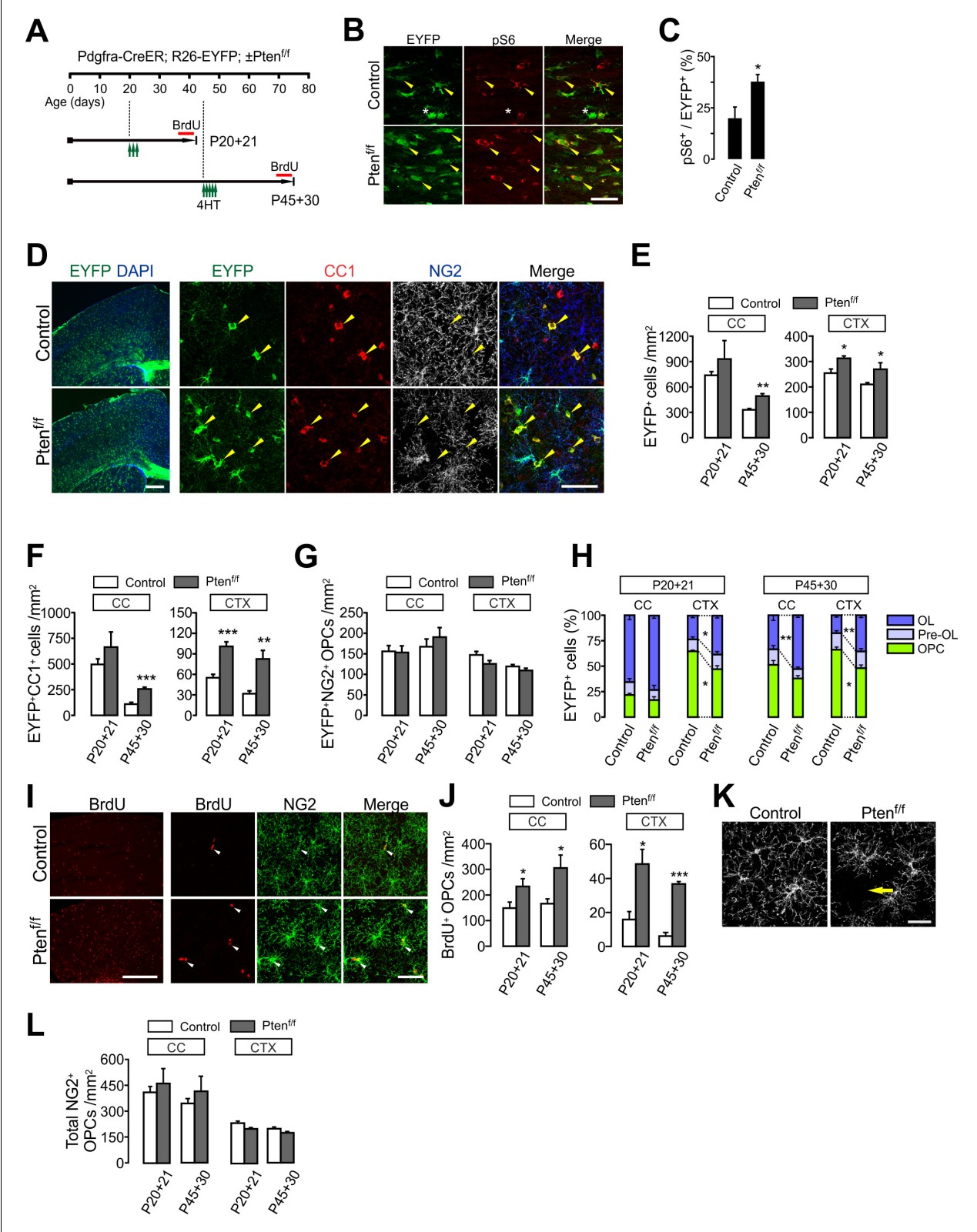

**Figure 2.** OPC-specific *Pten* ablation enhances oligodendrocyte differentiation and OPC proliferation in the brain. (A) Experimental scheme for 4HT injection and BrdU administration into *Pdgfra-CreER; R26-EYFP; ±Pten^{f/f} mice*, and for mouse sampling. For P45 +30, 4HT (1 mg per injection) was injected five times between P45 and P47 (a total of 5 mg). (B) Confocal images of phosphorylated S6 ribosomal protein (pS6) and EYFP⁺ cells in the CC at P20 +21. Arrowheads and asterisks indicate EYFP⁺ pS6⁺ cells and EYFP⁺pS6⁻ cells, respectively. Scale bar, 50 μm. (C) Percentage of pS6⁺ cells

*Figure 2 continued*

among EYFP-labeled cells in the CC at P20 +21. n = 5 mice per group. (**D**) Fluorescence (left) and confocal (right) images of EYFP$^+$ cells in the brains of the 4HT-administered control and *Pten* cKO mice (P20 +21). The confocal images were taken from the CTX. Arrowheads indicate EYFP$^+$CC1$^+$ mature OLs. Scale bars, 500 µm (left) and 50 µm (right). (**E**) Number of total EYFP$^+$ cells was increased in the CTX of *Pten* cKO mice during the OPC fate analysis for the two age windows. (**F**) Number of EYFP$^+$CC1$^+$ OLs. (**G**) The numbers of EYFP$^+$NG2$^+$ OPCs were not changed by the *Pten* cKO. (**H**) Percentages of OPC, pre-OL and OL among EYFP-labeled cells. (**I**) Fluorescence (left) and confocal (right) images of BrdU$^+$ cells in the CTX (P20 +21). Arrowheads indicate BrdU$^+$NG2$^+$ OPCs. Scale bars, 500 µm (left) or 50 µm (right). (**J**) Quantification of BrdU$^+$NG2$^+$ OPCs in the CC and CTX. (**K**) Confocal images showing disruption of tiled OPC distribution in the CTX of *Pten* cKO mice (P20 +21). An arrow indicates a cortical area devoid of an NG2$^+$ OPC. Scale bar, 50 µm. (**L**) Number of total OPCs. Data are represented as mean ±S.E.M. n = 4 ~ 7 mice per group for P20 ~41. n = 3 ~ 5 mice per group for P45 ~P75. *p<0.05; **p<0.01; ***p<0.001. Unpaired Student's t-test. The numerical data for the graphs are available in *Figure 2—source data 1*.

DOI: https://doi.org/10.7554/eLife.32021.007

The following source data and figure supplements are available for figure 2:

**Source data 1.** Numerical data for graphs in *Figure 2*.
DOI: https://doi.org/10.7554/eLife.32021.010

**Figure supplement 1.** OPC-specific *Pten* ablation enhances oligodendrocyte differentiation in the spinal cord.
DOI: https://doi.org/10.7554/eLife.32021.008

**Figure supplement 1—source data 1.** Numerical data for graphs in *Figure 2—figure supplement 1*.
DOI: https://doi.org/10.7554/eLife.32021.009

Thus, we asked whether PTEN-deficient OPCs continue to add new OLs, leading to an increase in total OLs. To quantify total OLs unambiguously, we used *Mobp-EGFP* BAC Tg mice in which EGFP is expressed only in mature OLs (*Kang et al., 2013*). *Pdgfra-CreER; Mobp-EGFP; ±Pten$^{f/f}$* mice received three 4HT injections (a total of 3 mg) between P14 and P18 (*Figure 3—figure supplement 1A*). At P30, there were increases in both EGFP$^+$ OLs (*Figure 3—figure supplement 1B,C*) and Olig2$^+$CC1$^+$ OLs (*Figure 3—figure supplement 1D,E*) in the CTX of *Pten* cKO mice.

For a longer-term OPC fate analysis, the 4HT-administered (at P20) *Pdgfra-CreER; R26-EYFP; ±Pten$^{f/f}$* mice were observed at P90 (thus P20 +70) (*Figure 3A*). These cKO mice exhibited even greater accumulation of EYFP$^+$ OLs in both the CC and CTX (a seven-fold increase of OL at P20 +70 vs. a two-fold increase at P20 +21) (*Figures 3B–D and and 2F*). Due to the greater new OL addition, total CC1$^+$ OLs also increased (*Figure 3E*). Interestingly, EYFP$^+$NG2$^+$ OPCs were significantly reduced in the CC (p=0.0004), but not in the CTX of *Pten* cKO mice (P20 +70) (*Figure 3F*). Due to the marked EYFP$^+$ OL accumulation, the percentages of mature OLs among EYFP-labeled cells were increased, both in the CC and CTX, whereas those of labeled OPCs were highly reduced (*Figure 3G*). It is likely that the PTEN-deficient (EYFP$^+$) OPCs in the CC may have undergone OL differentiation more rapidly than their own cell division, and thus EYFP-labeled OPCs markedly decreased at P20 +70. At the same time, PTEN-harboring EYFP$^-$ OPCs may have replaced the EYFP$^+$ OPCs. Indeed, at P20 +70, EYFP$^-$Olig2$^+$NG2$^+$ OPCs were predominant in the CC of *Pten* cKO mice (*Figure 3H*), and thus, the density of total OPCs was not altered (*Figure 3I*).

BrdU administration (50 mg/kg per injection, a total of 10 injections between P80 and P84) and cell proliferation analyses revealed that OPCs continuously proliferate at higher rates in the CTX of *Pten* cKO (*Figure 3J,K*), and yet without changes in total OPC number (*Figure 3I*). Some of the BrdU-laden OPCs appeared to have become pre-OLs or OLs (EYFP$^+$NG2$^-$) during the last week before sampling (*Figure 3K* right).

These results further confirm that PTEN regulates OPC differentiation, not only in early postnatal life but also at later adult ages. These also suggest that the enhanced OL differentiation by PTEN inactivation is not a transient event, and that the newly generated OLs from PTEN-deficient OPCs survive long. Although OPC proliferation remained promoted in the CTX, it appears that this enhanced proliferation is tightly controlled to maintain OPC homeostasis.

Although in sharp contrast to earlier studies in which no changes (*Goebbels et al., 2010; Harrington et al., 2010*) or a decrease (*Maire et al., 2014*) in OL numbers were observed with different Cre-dependent *Pten* deletion, our results provide an important example of how the outcomes of OPC-specific genetic manipulations can differ from those of mature OL-specific or non-cell-specific cKO when assessing cell-intrinsic regulators for adult OL development.

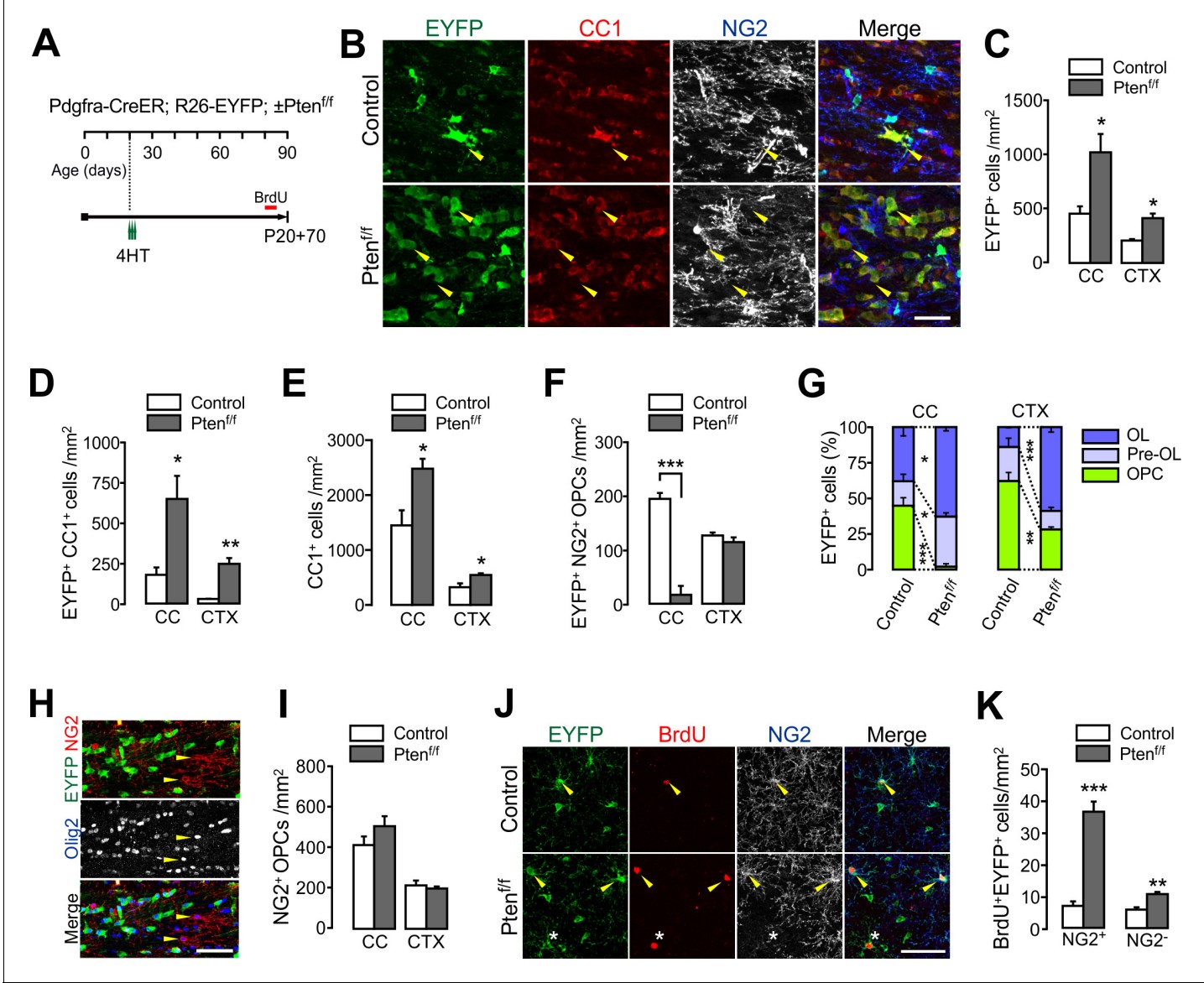

**Figure 3.** PTEN-deficient OPCs continuously add new oligodendrocytes. (**A**) Experimental scheme for 4HT and BrdU administration into *Pdgfra-CreER; R26-EYFP; ±Pten^f/f* mice, and for mouse sampling at P90 (P20 +70). (**B**) Confocal images of EYFP+ cells in the CC showing their maturation stage. Arrowheads indicate EYFP+CC1+ OLs. Scale bar, 50 μm. (**C**) Number of EYFP+ cells in the CC and CTX at P20 +70. (**D**) Number of EYFP+CC1+ OLs. (**E**) Number of total CC1+ OLs. (**F**) Number of EYFP+NG2+ OPCs. (**G**) Percentages of OPC, pre-OL and OL among EYFP-labeled cells in the control and *Pten* cKO mice at P20 +70. (**H**) Confocal images of NG2+ OPCs in the CC of *Pten* cKO mice. At P20 +70, resident Olig2+NG2+ OPCs did not EYFP+ OPCs (arrowheads) in the CC. Scale bar, 50 μm. (**I**) The number of total NG2+ OPCs did not change in *Pten* cKO mice. (**J**) Confocal images of cortical EYFP-labeled BrdU+ cells. Arrowheads and asterisk indicate BrdU+NG2+ OPCs and a BrdU+NG2- cell, respectively. (**K**) Number of BrdU+EYFP+NG2+ OPCs and BrdU+EYFP+NG2- OL lineage cells in the CTX. Data are represented as mean ±S.E.M. n = 3 (control) and 4 (*Pten* cKO) mice. *p<0.05; **p<0.01; ***p<0.001. Unpaired Student's t-test. The numerical data for the graphs are available in *Figure 3—source data 1*.

DOI: https://doi.org/10.7554/eLife.32021.011

The following source data and figure supplements are available for figure 3:

**Source data 1.** Numerical data for graphs in *Figure 3*.
DOI: https://doi.org/10.7554/eLife.32021.014

**Figure supplement 1.** OPC-specific *Pten* ablation facilitates oligodendrocyte accumulation in the cortex.
DOI: https://doi.org/10.7554/eLife.32021.012

**Figure supplement 1—source data 1.** Numerical data for graphs in *Figure 3—figure supplement 1*.
DOI: https://doi.org/10.7554/eLife.32021.013

## OPC-specific *Pten* ablation promotes new axon myelination

In order to determine whether the increased new OLs contribute to enhanced myelination in *Pten* cKO mice, we performed western blot analysis for myelin proteins. 4HT was injected into *Pten*^f/f (control) or *Pdgfra-CreER; Pten*^f/f mice four times between P14 and P21 (a total of 4 mg) (*Figure 4A*). Even though *Pten* might have been deleted from only subsets of young OPCs by this Cre activation protocol, levels of PLP, MOG, and CNP were significantly increased in the CTX of *Pten* cKO mice at P75 (*Figure 4B,C*). Moreover, electron microscopic (EM) analyses revealed that the *g*-ratios (the ratio between the inner and the outer diameters of the myelin sheath) of callosal axon fibers were significantly lowered in the *Pten* cKO mice (P13 +17) (p=0.038) (*Figure 4D–F*). More importantly, there was a marked increase in the number of myelinated axons (p=0.033) (*Figure 4G*), suggesting that the increased number of new OLs myelinate more axons in OPC-specific *Pten* cKO mice.

To better observe the growth of OL processes, *Pdgfra-CreER;* ±*Pten*^f/f mice were crossed with *R26-mEGFP (mT/mG)* mice, which express the membrane-bound EGFP upon Cre recombination (*Muzumdar et al., 2007*). In a few weeks after 4HT administration (at P25), EGFP^+ cells exhibited more rapid morphological changes in the *Pten* cKO mice, and the EGFP^+ OPC progeny cells formed more thin processes along axonal tracts (*Figure 4H,I*). Consequently, at P25 +20, discrete stellate-shaped OPC morphology was no longer seen in the CC of cKO mice (*Figure 4H*). We also performed immuno-EM with *Pdgfra-CreER; R26-mEGFP;* ±*Pten*^f/f (P20 +21) mice and anti-EGFP antibodies, in which the gold particles labeled only EGFP^+ membranes, including newly formed myelin sheaths (*Figure 4J*). The immuno-EM analysis revealed that the percentage of newly formed myelinated axons increased by about 2.5-fold in the CC of the *Pten* cKO mice (p=0.0179) (*Figure 4K*). These results suggest that PTEN inactivation in OPCs facilitates and enhances new myelination.

## OL-specific *Pten* ablation does not change OL number, but does increase myelin thickness

In two earlier independent studies, Akt signaling was activated in OLs by using *Plp1-Akt1 (DD)* transgenic mice (*Flores et al., 2008*) or *Cnp-Cre; Pten*^f/f mice (*Goebbels et al., 2010*). Both mutant mice exhibited similar phenotypes, such as thicker myelin sheaths, but with no change in OL number and OL regeneration. However, the gene promoters utilized (i.e. *Plp1* and *Cnp*) in those studies may have affected at most only a small fraction of OPCs, given their endogenous gene expression patterns. Thus, the phenotypic differences between our results and those earlier ones may have stemmed from different target cell populations (i.e. OPC vs. OL). Alternatively, it is also possible that different methods utilized in our OL lineage assessments caused these differences. To determine whether differences in target cell populations for PTEN-Akt manipulation account for the observed phenotypic differences, we used *Mog-iCre* mice (*Buch et al., 2005*) for *Pten* deletion. In *Mog-iCre; R26-EYFP;* ±*Pten*^f/f mice (P41), EYFP was expressed only in CC1^+ OLs (*Figure 5A*), suggesting that the Cre activity is highly specific to OLs. The numbers of EYFP^+ OLs and NG2^+ OPCs were not changed by the *Mog-iCre*-based *Pten* cKO for both the brain and the spinal cord (*Figure 5B,C*). Moreover, EM analysis of *R26-EYFP; Pten*^f/f; ±*Mog-iCre* (P34) mice revealed that myelin sheaths were thickened (*Figure 5D–F*), but the percentage of myelinated axons did not increase in the cKO mice (*Figure 5G*). These results are in accordance with the earlier reports (*Flores et al., 2008*; *Goebbels et al., 2010*), and together they support the notion that the enhanced oligodendrogenesis (promoted OL generation and increase in myelinated axons) in *Pdgfra-CreER; Pten*^f/f mice is due to OPC-specific targeting of *Pten*, thus suggesting differential cell-stage-specific roles of PTEN.

## PTEN-deficient OPCs rapidly regenerate OLs after lysolecithin-induced demyelination

Given the facilitated OL development in OPC-specific *Pten* cKO mice, we asked whether *Pten*-inactivated OPCs also enhance OL regeneration. After *Pdgfra-CreER; R26-EYFP;* ±*Pten*^f/f mice received tamoxifen (a total of 10 injections for 5 days) from P50 to P54, 1% lysolecithin (LPC) was injected into the CC at P58 (*Figure 6A*). This LPC injection typically induces focal demyelination in 3 days (*Figure 6B*), and OL regeneration (and remyelination) occurs for the following 3 weeks. Two weeks after LPC, we observed that more EYFP^+CC1^+ OLs accumulated in the CC of *Pten* cKO mice (p=0.01) (*Figure 6C–E*), while EYFP-labeled OPCs were unchanged (*Figure 6F*). Thus, the

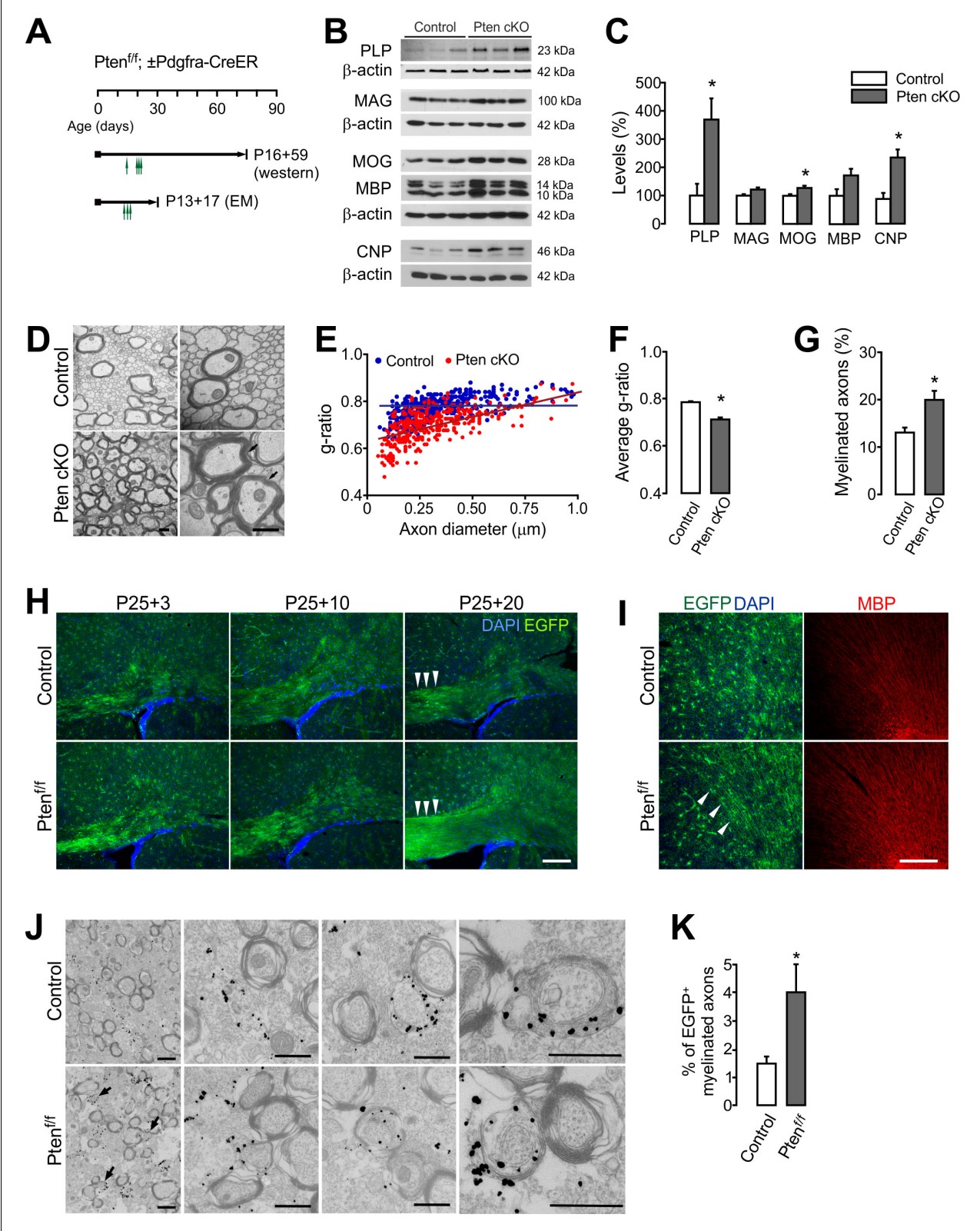

**Figure 4.** OPC-specific *Pten* ablation promotes new myelination. (**A**) Experimental scheme for 4HT into *Pten^{f/f}* ± *Pdgfra-CreER* mice, and for mouse sampling for western blot analysis (**B,C**) and EM analyses (D –G). For western blot analysis, 4HT was injected into *Pten^{f/f}* (control) and *Pdgfra-CreER; Pten^{f/f}* mice at P14, P20 (twice), and P21 (1 mg per injection, a total of 4 mg), and the mice were sampled at P75. For EM, A single dose 4HT was injected at P13, 15, and 17 (a total of 3 mg), and the mice were sampled at P30. (**B**) Western blot analysis of cortical lysates for myelin proteins of
*Figure 4 continued on next page*

*Figure 4 continued*

control and OPC-targeted *Pten* cKO mice (P14 +61). (**C**) Quantification of the levels of PLP, MAG, MOG, MBP and CNP in the western blot (**B**). n = 3 mice per group. (**D**) Representative electron micrograph (EM) of the CC of the control and *Pten* cKO mice (P13 +17). Arrows indicate thickened myelin in *Pten* cKO mice. Scale bar, 500 nm. (**E**) Scatter plot of g-ratios. More than 100 myelinated axons per mouse were analyzed. (**F**) Average g-ratio. (**G**) Percentage of myelinated axons was increased in *Pten* cKO mice. More than 700 axons were analyzed for myelination per mouse. n = 3 mice per group for (D - G). (**H, I**) Fluorescence images of EGFP$^+$ cells in the CC (**G**) and CTX (**H**) of *Pdgfra-CreER; R26-mEGFP;±Pten$^{f/f}$* mice. 4HT (1 mg per injection, two injections per day) was injected at P25 and P26 (a total of 4 mg), and the mice were killed 3, 10, or 20 days later. Arrowheads indicate increased EGFP$^+$ slender processes, reminiscent of bundles of myelinated fibers in *Pten* cKO mice (P25 +20). Scale bar, 100 μm. (**J**) Immuno-EM of anti-EGFP immuno-gold particles in the CC of *Pdgfra-CreER; R26-mEGFP; ±Pten$^{f/f}$* mice (P20 +21). Arrows indicate EGFP$^+$ newly formed immature myelin sheaths. Scale bar, 500 nm. (**K**) Percentage of EGFP$^+$ myelinated axons increased at P20 +21. n = 3 mice per group. Data are represented as mean ±S.E.M. *p<0.05. Unpaired Student's t-test. The numerical data for the graphs are available in *Figure 4—source data 1*. Original western images are available in *Figure 4—source data 2*.

DOI: https://doi.org/10.7554/eLife.32021.015

The following source data is available for figure 4:

**Source data 1.** Numerical data for graphs in *Figure 4*.
DOI: https://doi.org/10.7554/eLife.32021.016
**Source data 2.** Original western blot images used for *Figure 4A*.
DOI: https://doi.org/10.7554/eLife.32021.017

percentage composition of EYFP-labeled cells changed, which indicates more OL generation from OPCs (*Figure 6G*). EM analyses also revealed that myelin thickness was increased, particularly for large-diameter axons (diameters > 1 μm, p=0.0458) in the *Pten* cKO mice (*Figure 6H–K*).

We also injected LPC into the dorsal SC WM (*Figure 6—figure supplement 1A,B*). To this cohort of mice, BrdU was also administered from 1 dpi (1 day after LPC injection) until mouse sampling (*Figure 6—figure supplement 1A*). We observed that more EYFP$^+$CC1$^+$ OLs accumulated in the lesions of *Pten* cKO than in the controls (p=0.0008) (*Figure 6—figure supplement 1C,D*). Enhanced OL accumulation in the cKO mice was also evident based on the number of BrdU$^+$CC1$^+$ cells (*Figure 6—figure supplement 1D*), indicating that the new OLs had been generated from proliferating OPCs.

Together, these results suggest that *Pten*-inactivated OPCs have a greater potential to regenerate new OLs than normal OPCs in the demyelinated CNS.

## mTOR signaling is not required for the enhanced OL development from PTEN-inactivated OPCs

Although PI3K-Akt potentially regulates numerous downstream signaling cascades, mTOR is often thought to be the main downstream effector in OL development and myelination (*Figlia et al., 2018*; *Wahl et al., 2014*; *Wood et al., 2013*). However, our OPC-specific *Tsc1* ablation (inducing mTORC1 hyperactivity) impaired OL development, whereas OPC-specific *Pten*-ablation (which also activates mTORC1) promoted new OL generation. Thus, these opposite results raise the possibility that the *Pten*-deletion-induced OL promotion is not mediated only by mTORC1 activity. While mTORC1 signaling is important for myelin growth, it is also possible that different PTEN-related signaling mechanisms regulate the OPC-to-OL lineage progression. Indeed, we observed that almost all pS6 (an mTORC1 activity indicator) immunoreactivities were localized at either EYFP$^+$NG2$^-$ pre-OLs (about 25%) or EYFP$^+$CC1$^+$ OLs (more than 70%) in the CC of *Pdgfra-CreER; R26-EYFP; ±Pten$^{f/f}$* (P20 +21) mice. The percentage of NG2$^+$ OPCs among pS6$^+$ cells was negligible (less than 1%) even after *Pten* cKO (*Figure 7A*). These observations suggest that the actual site of mTORC1 activity of OL lineage is not NG2$^+$ OPC, but more mature OL lineage cells.

In order to address the question of whether mTOR-dependent signaling mediates the enhanced OL generation from PTEN-ablated OPCs, we deleted both *Mtor* and *Pten* from OPCs with *Pdgfra-CreER; R26-EYFP; ±Pten$^{f/f}$; ±Mtor$^{f/f}$* (*Risson et al., 2009*) mice at P20. First, the effectiveness of *Mtor* deletion was confirmed by a marked decrease of EYFP$^+$pS6$^+$ cells in *Mtor* or *Pten-Mtor* double cKO mice (*Figure 7B,C*). To our surprise, the degree of enhancement of OL generation was comparable between *Pten* cKO and *Pten-Mtor* double cKO mice (P20 +21) (*Figure 7—figure supplement 1*), indicating that the *Pten* cKO phenotypes were not reversed, even after mTOR was effectively inactivated.

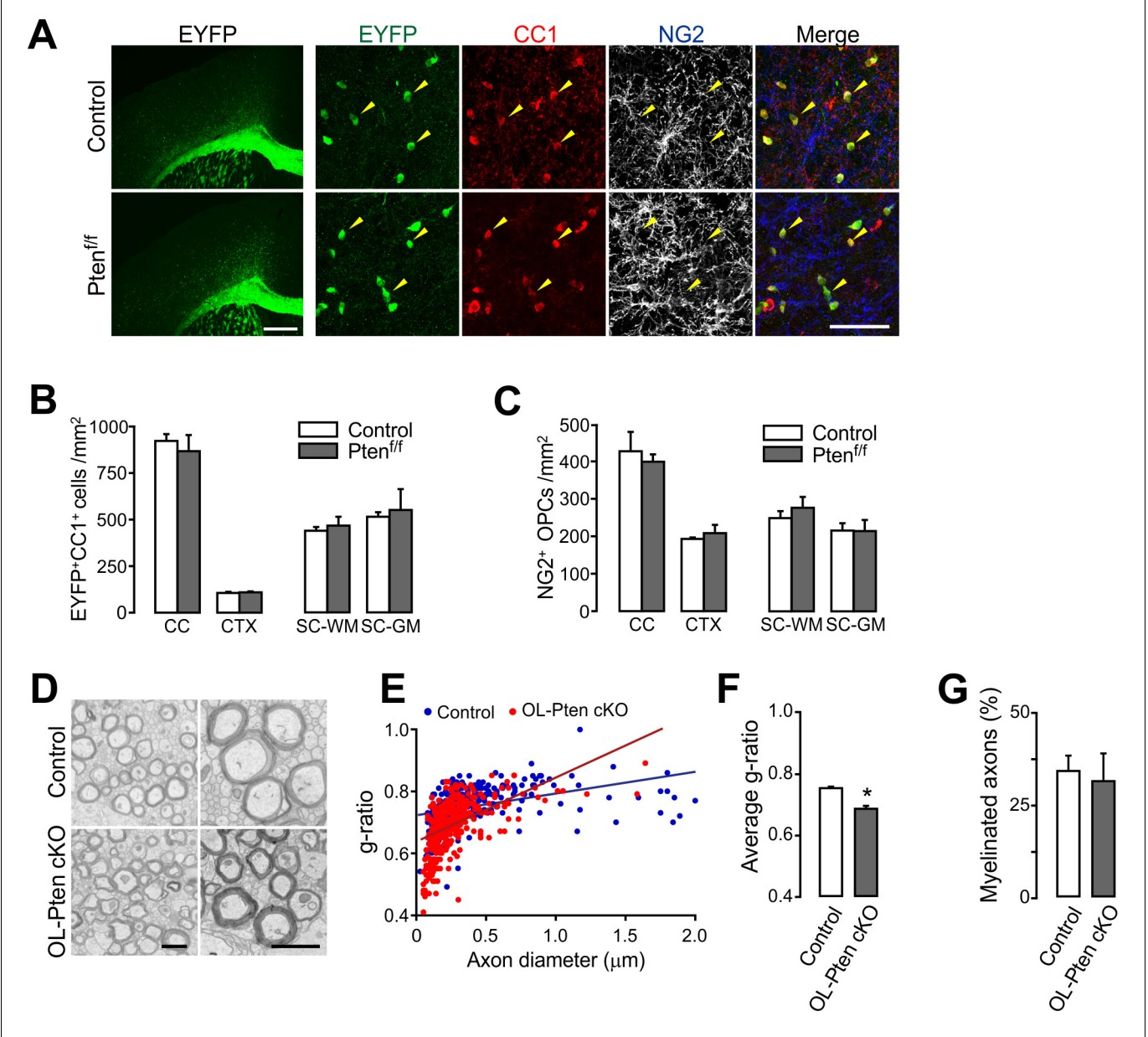

**Figure 5.** Oligodendrocyte numbers and the degree of new myelination are not changed by OL-specific *Pten* ablation. (A) Fluorescence (left) and confocal (right) images of EYFP$^+$ cells in *Mog-iCre; R26-EYFP; ±Pten$^{f/f}$* (P41) mice. The confocal images were taken from the CTX. Arrowheads indicate EYFP$^+$CC1$^+$ OLs. Scale bars, 500 µm (left) and 50 (right) µm. (B) Number of EYFP$^+$CC1$^+$ OLs in the brain and the SC. (C) Number of NG2$^+$ OPCs. (D) Representative EM of myelinated callosal axons from *R26-EYFP; Pten$^{f/f}$* (control) and *R26-EYFP; Pten$^{f/f}$; ±Mog-iCre* (*Pten* cKO) mice (P34). Scale bar, 500 nm. (E) Scatter plot of the g-ratios. More than 100 axons per mouse, and three mice per group were analyzed. (F) Average g-ratio. (G) Percentage of myelinated axons was not altered in OL-specific *Pten* cKO mice. Data are represented as mean ±S.E.M. n = 3 (control) or 4 (*Pten* cKO) mice for (B, C), and n = 3 mice per group for (D - G). *p<0.05. Unpaired Student's t-test. The numerical data for the graphs are available in *Figure 5—source data 1*.
DOI: https://doi.org/10.7554/eLife.32021.018

The following source data is available for figure 5:

**Source data 1.** Numerical data for graphs in *Figure 5*.
DOI: https://doi.org/10.7554/eLife.32021.019

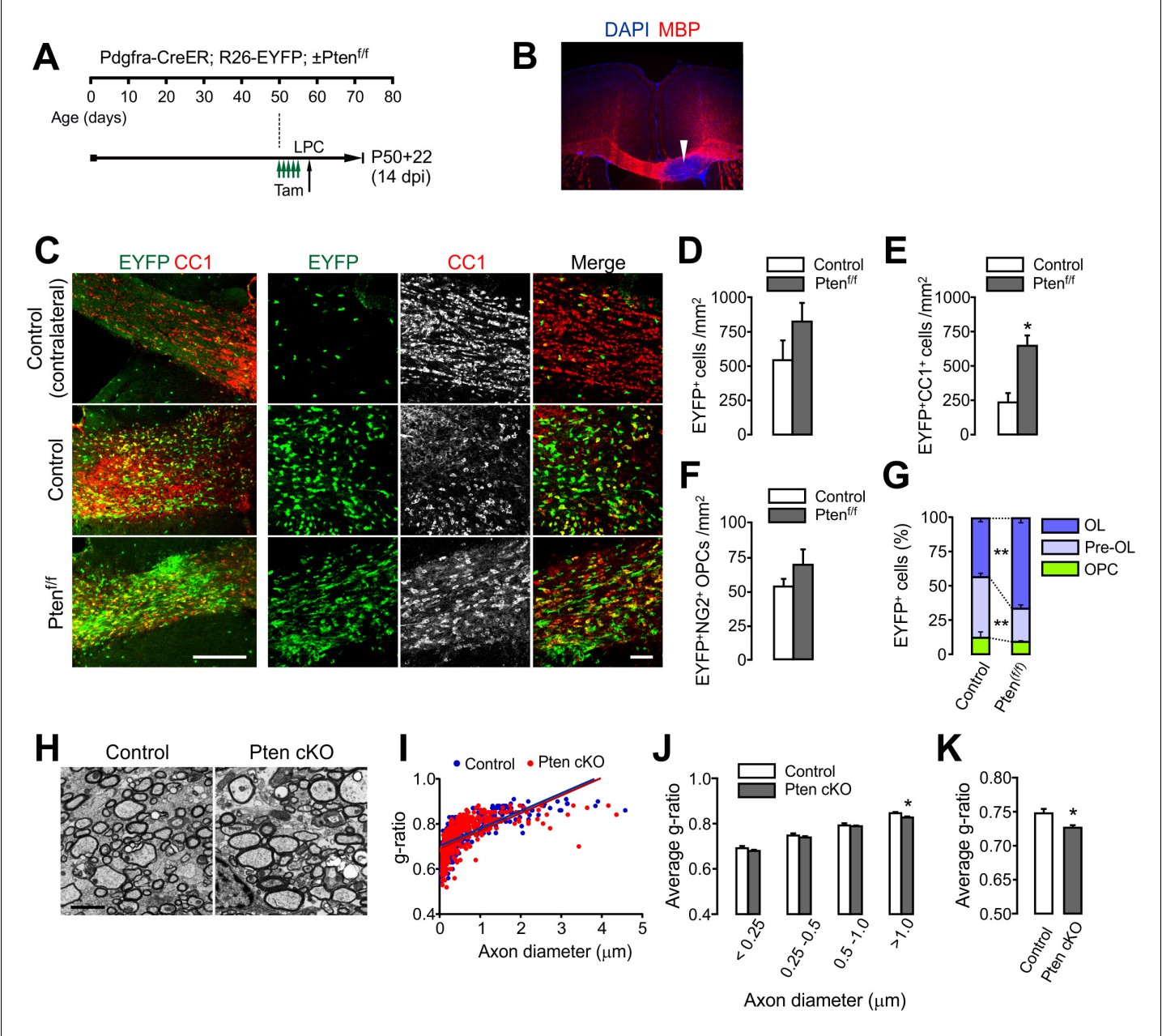

**Figure 6.** OPC-specific *Pten* ablation facilitates oligodendrocyte regeneration and remyelination after LPC-induced demyelination. (A) Experimental scheme for tamoxifen (Tam) and lysolecithin (LPC) injection into *Pdgfra-CreER; R26-EYFP; ±Pten*[f/f] mice. A series of tamoxifen injections (40 mg/kg per i.p. injection, a total of 10 injections) were given between P50 and P54. The demyelination was induced with LPC injection at P58, and the mice were sampled at P72, which is 14 days after LPC injection (14 dpi), and 22 days after the first tamoxifen injection (P50 +22). (B) Loss of MBP immunoreactivity (arrowhead) at the LPC-injected CC in the brain. (C) Fluorescence (left) and confocal (right) images of EYFP[+] cells and CC1[+] OLs in the CC at 14 days after LPC injection (14 dpi). Scale bars, 200 μm (left) and 50 μm (right). (D) Quantification of EYFP[+] cells in the LPC injected site at 14 dpi. (E) Quantification of EYFP[+]CC1[+] OLs. (F) Quantification of EYFP[+]NG2[+] OPCs. (G) *Pten* cKO mice exhibited marked changes in the percentage of OL and pre-OL among EYFP-labeled cells at the LPC-injected areas at 14 dpi. (H) Representative EMs of the lesion in the CC at 14 dpi. Scale bar, 2 μm. (I) Scatter plot for individual g-ratios of myelinated axons. More than 100 myelinated axons per mouse, and three mice per group were analyzed. (J, K) Average g-ratio according to axon diameter (J) and as a total (K). Data are represented as mean ±S.E.M. n = 3 (control) or 5 (*Pten* cKO) mice for (D - G), or n = 3 per group for (H–K). *p<0.05; **p<0.01. Unpaired Student's t-test. The numerical data for the graphs are available in *Figure 6—source data 1*.
DOI: https://doi.org/10.7554/eLife.32021.020

The following source data and figure supplements are available for figure 6:

**Source data 1.** Numerical data for graphs in *Figure 6*.

*Figure 6 continued*

DOI: https://doi.org/10.7554/eLife.32021.023

**Figure supplement 1.** OPC-specific *Pten* ablation facilitates oligodendrocyte regeneration after lysolecithin-induced demyelination in the spinal cord.

DOI: https://doi.org/10.7554/eLife.32021.021

**Figure supplement 1—source data 1.** Numerical data for graphs in *Figure 6—figure supplement 1*.

DOI: https://doi.org/10.7554/eLife.32021.022

On the other hand, after OPC-specific *Mtor* deletion in 4HT-administered *Pdgfra-CreER; R26-EYFP; Mtor^{f/f}* mice (P20 +21), there was a marked reduction in EYFP$^+$ cells compared to controls, particularly for CC1$^+$ OLs, but not NG2$^+$ OPCs (*Figure 7D–G*), confirming a critical role of mTOR in OL development (*Wahl et al., 2014*). However, most strikingly, new OL (EYFP$^+$CC1$^+$) generation was still greatly enhanced in the OPC-targeted *Pten-Mtor* double cKO mice, as compared to *Mtor* cKO mice (*Figure 7D–F*). Consequently, even without mTOR, the *Pten* deletion sufficiently increased the percentage of mature OLs among EYFP-labeled cells in the CC and CTX (*Figure 7H*). The *Pten*-deletion-induced OL promotion was also observed in the ventral GM area of the SC of *Pten-Mtor* cKO mice: OPC-specific *Mtor* cKO reduced new OL accumulation at P20 +21, but *Pten-Mtor* double cKO reversed this phenotype, or showed even more new OLs than in controls (*Figure 7—figure supplement 2*).

EM analyses confirmed that OPC-specific *Pten-Mtor* double cKO lowered the *g*-ratio to a level comparable to *Pten* cKO (*Figure 7I–K*), and formed new myelination for more axons (*Figure 7L*). These striking results suggest that one or more alternative signaling mechanisms to mTOR mediate the enhanced OL differentiation after OPC-specific PTEN inactivation.

## PTEN ablation induces the inhibitory phosphorylation of GSK3β in OPCs

GSK3β is another downstream target of activated Akt (*Figure 8A*), and it regulates cell differentiation, neural progenitor proliferation, and axonal growth in the CNS (*Hur and Zhou, 2010*). Interestingly, when cultured OPCs were treated by triiodothyronine (T3, 30 ng/ml) to induce OL differentiation, the levels of phospho-GSK3β$^{Ser9}$ were significantly increased by ~65% in 1 day (p=0.028, paired t-test), at which point the cultured cells are a mixture of pre-OLs (GalC$^+$,O4$^+$, MBP$^-$) (*Dai et al., 2014*) and a small fraction of OPCs (weak PDGFRα$^+$) (*Figure 8B–D*).

Although Akt-mediated phosphorylation at Ser9 inhibits GSK3β activity (*Cross et al., 1995*), it has not been shown that PTEN ablation leads to increased p-GSK3β$^{Ser9}$ levels in OL lineage cells (*Goebbels et al., 2017*). To determine whether PTEN ablation results in an increase of the inhibitory phosphorylation of GSK3β in OPCs, we isolated *Pten*-deleted EYFP$^+$ cells from the *Pdgfra-CreER; R26-EYFP; ±Pten^{f/f}* mice (P20 +21) by fluorescence-activated cell sorting (FACS). Western blot analysis of the isolated EYFP$^+$ cells revealed that the levels of p-GSK3β$^{Ser9}$ were significantly increased by PTEN ablation (p=0.0138) (*Figure 8E*). It also appeared that there were increases in the levels of phosphorylated Akt (p-Akt$^{Ser473}$) and p-GSK3β$^{Ser9}$ in the CTX of *Olig1-Cre; Pten^{f/f}* late (E18.5) embryos, compared with the *Olig1-Cre* embryos (*Figure 8F*). These observations provide compelling evidence that PTEN inactivation in OPCs leads to GSK3β inactivation *in vivo*. Moreover, in our OPC culture from the 4HT-administered *Pten^{f/f}; ±Pdgfra-CreER* (P1 +1) pups, there was a consistent tendency (but not significant, n = 3 replicates) of increase of p-Akt$^{Ser473}$ (*Figure 8G*) and p-GSK3β$^{Ser9}$ (*Figure 8H*) in *Pten* cKO OPCs.

## OPC-targeted GSK3β inactivation enhances OL development and new myelination

Next, we examined the effects of OPC-specific GSK3β inactivation *in vivo* as a likely signaling event after PTEN ablation. We bred *Pdgfra-CreER; R26-EYFP* mice with *Gsk3b^{f/f}* mice (*Patel et al., 2008*), and injected tamoxifen into *Pdgfra-CreER; R26-EYFP; ±Gsk3b^{f/f}* mice between P20 and P22 (*Figure 9A*). We isolated EYFP$^+$ cells from these mice at P20 +8, and performed RT-qPCR for *Gsk3b* mRNA levels. The results of RT-qPCR showed that there was a reduction of *Gsk3b* mRNA by more than 90% in EYFP$^+$ cells, confirming an effective *Gsk3b* ablation in OPCs (*Figure 9B*).

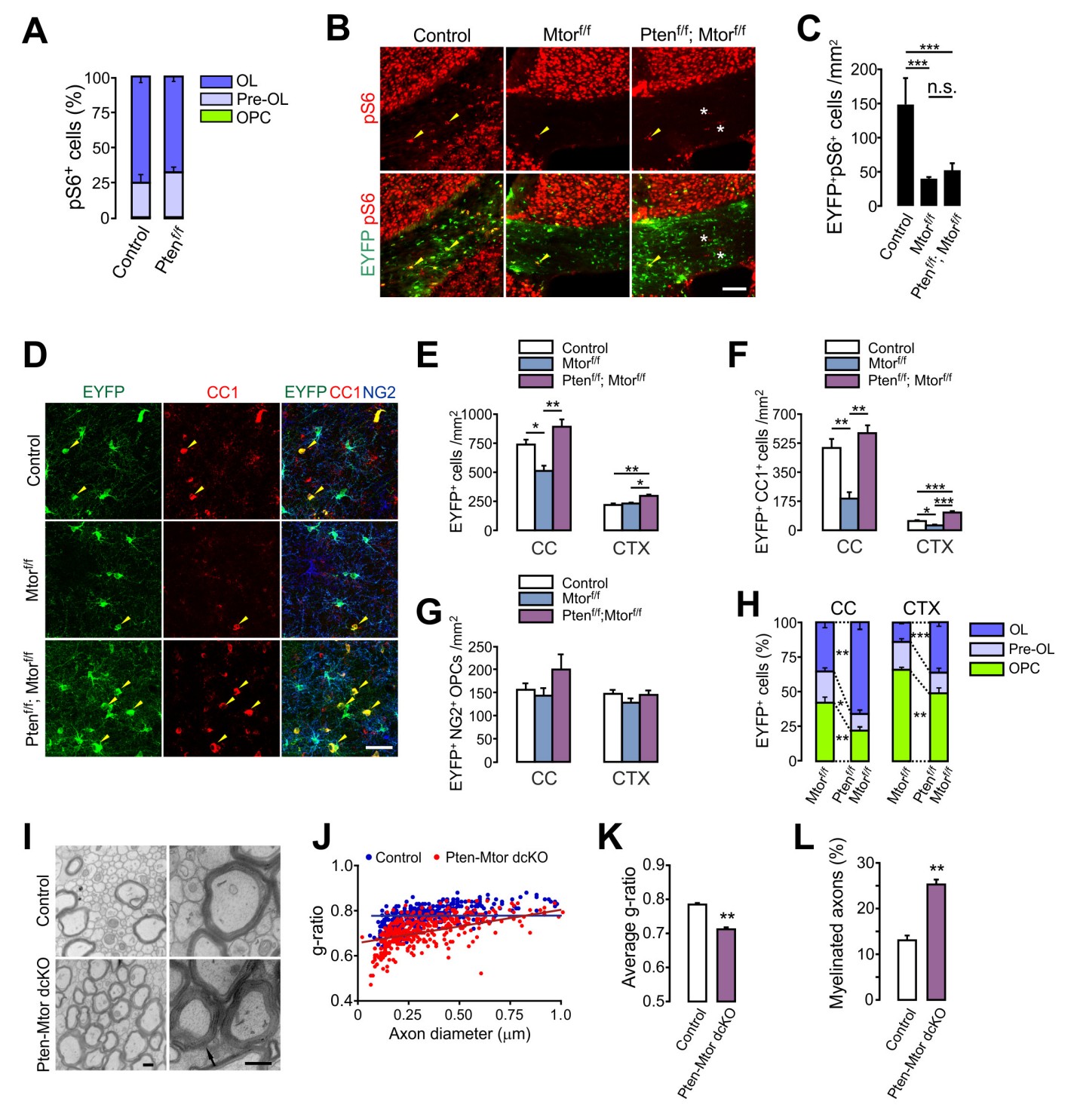

**Figure 7.** mTOR is dispensable for the enhanced oligodendrocyte generation from PTEN-deficient OPCs and the subsequent hypermyelination. (**A**) Percentage of OPC, pre-OL and OL among EYFP⁺pS6⁺ cells in the CC of the control (*Pdgfra-CreER; R26-EYFP*) and *Pten* cKO (*Pdgfra-CreER; R26-EYFP; Pten^f/f*) mice (P20 +21). Almost all pS6 immunoreactivities were observed either in EYFP⁺NG2⁻ pre-OLs or in EYFP⁺CC1⁺ OLs, but not in NG2⁺ OPCs. More than 150 callosal EYFP⁺pS6⁺ cells were analyzed per mouse. n = 4 mice per group. (**B**) Validation of effective mTOR inactivation with pS6 immunofluorescence in OPC-specific *Mtor* and *Pten-Mtor* double cKO mice (*Pdgfra-CreER; R26-EYFP; ±Pten^f/f; ±Mtor^f/f*, P20 +21). Arrowheads and asterisks indicate pS6⁺EYFP⁺ and pS6⁺EYFP⁻ cells, respectively. Scale bar, 100 μm. (**C**) Quantification of the EYFP⁺pS6⁺ cells in the CC. n = 3 mice per group. (**D**) Confocal images of EYFP⁺ cells in the CTX showing their maturation stage. Arrowheads indicate newly differentiated EYFP⁺CC1⁺ OLs. Scale bar, 50 μm. (**E - G**) Quantification of total EYFP⁺ cells (**E**), EYFP⁺CC1⁺ OLs (**F**), and EYFP⁺NG2⁺ OPCs (**G**) in the CC. (**H**) Percentages of OPC, pre-OL

*Figure 7 continued on next page*

*Figure 7 continued*

and OL among EYFP-labeled cells of *Mtor* cKO and *Pten-Mtor* cKO mice (P20 +21). (**I**) Representative EM of callosal axons in the control (*Pten^f/f*) and *Pten-Mtor* cKO mice (P13 +17). An arrow indicates altered myelin thickness. Scale bar, 500 nm. (**J**) Scatter plot of individual *g*-ratios. (**K**) Average of *g*-ratios. At least 130 axons per mouse, and three mice per group were analyzed. (**L**) Percentage of myelinated axons. Data are represented as mean ±S.E.M. n = 6 (control), 4 (*Mtor* cKO), or 4 (*Pten-Mtor* cKO) mice for (E - H), n = 3 mice per group for (I - L). *p<0.05; **p<0.01; ***p<0.001. One-way ANOVA with Bonferroni test for (C, E – G). Unpaired Student's *t*-test for (H, K, L). The numerical data for the graphs are available in *Figure 7—source data 1*.
DOI: https://doi.org/10.7554/eLife.32021.024

The following source data and figure supplements are available for figure 7:

**Source data 1.** Numerical data for graphs in *Figure 7*.
DOI: https://doi.org/10.7554/eLife.32021.029

**Figure supplement 1.** Additional *Mtor* ablation does not reverse *Pten*-ablation-induced oligodendrocyte promotion in the brain.
DOI: https://doi.org/10.7554/eLife.32021.025

**Figure supplement 1—source data 1.** Numerical data for graphs in *Figure 7—figure supplement 1*.
DOI: https://doi.org/10.7554/eLife.32021.026

**Figure supplement 2.** Additional *Pten* ablation restores oligodendrocyte differentiation in the spinal cord of *Mtor* cKO mice.
DOI: https://doi.org/10.7554/eLife.32021.027

**Figure supplement 2—source data 1.** Numerical data for graphs in *Figure 7—figure supplement 2*.
DOI: https://doi.org/10.7554/eLife.32021.028

The tamoxifen-administered *Pdgfra-CreER; R26-EYFP; ±Gsk3b^f/f* were also examined 21 days later (P20 +21). The last 4-day-BrdU chasing (see *Figure 9A*) revealed significantly enhanced OPC proliferation in the *Gsk3b* cKO mice, which is similar to the OPC-specific *Pten* cKO mice (*Figure 9C*). Moreover, there were marked increases in both total EYFP⁺ cells and EYFP⁺CC1⁺ OLs in both the CC and CTX of *Gsk3b* cKO mice (*Figure 9D–F*), suggesting enhanced OL generation. In contrast, densities of EYFP⁺NG2⁺ OPCs and total OPCs were not altered (*Figure 9G,H*), and as a consequence, the percentage of OLs or that of OPCs among EYFP⁺ OL lineage cells was altered (*Figure 9I*). EM analyses also confirmed that myelin thickness and the degree of new axon myelination were significantly enhanced by *Gsk3b* cKO (P13 +17) (*Figure 9J–M*). These results strongly suggest that resting GSK3β activity inhibits OPC-to-OL lineage progression in the brain, and similar to PTEN ablation, OPC-specific GSK3β inactivation promotes OL development.

We also targeted *Gsk3b* specifically in OLs with *Mog-iCre; ±Gsk3b^f/f* mice (P31). Similar to OL-specific *Pten* cKO mice, there was no change in numbers of CC1⁺ OL and OPCs (*Figure 9—figure supplement 1A,B*), but the levels of MBP and CNP were increased in the OL-specific *Gsk3b* cKO mice (*Figure 9—figure supplement 1C,D*). These results suggest that GSK3β, like PTEN, negatively regulates not only OPC-to-OL transition but also myelin protein synthesis.

## Discussion

PI3K-Akt-mTORC1 signaling is critical for early myelination (*Figlia et al., 2018*; *Wood et al., 2013*), but precise control of mTORC1 activity is highly important for the progression of cell maturation and myelination. The results from OPC-specific*Tsc1* cKO mice in this study and other previous results (*Jiang et al., 2016*; *Lebrun-Julien et al., 2014*) suggest that mTORC1 hyperactivity either causes OL lineage arrest or is detrimental to OL survival. Similarly, *Figlia et al. (2017)* showed that *Tsc1* cKO-induced mTORC1 hyperactivity delays the onset of Schwann cell myelination in the PNS, although residual mTORC1 activity derives myelin growth at later stages. Therefore, a large body of evidence suggests that even though mTORC1 is necessary, uncontrolled stimulation of mTORC1 activity may not be a practical approach for adult myelin repair.

This study provides the first *in vivo* genetic evidence that, in parallel with PI3K-Akt-mTORC1 signaling, PTEN-Akt-GSK3β signaling negatively regulates OL development and myelination. In particular, we demonstrate that PTEN-dependent inhibition of OPC-to-OL progression widely occurs both in the brain and the SC, and is persistent into adulthood. Our findings reveal that inactivation of PTEN in OPCs mobilizes its cell-intrinsic downstream signaling via mTOR-independent mechanisms, probably by controlling GSK3β activity. Thus, OL regeneration may be effectively enhanced by OPC-targeted inactivation of PTEN or GSK3β in the adult CNS.

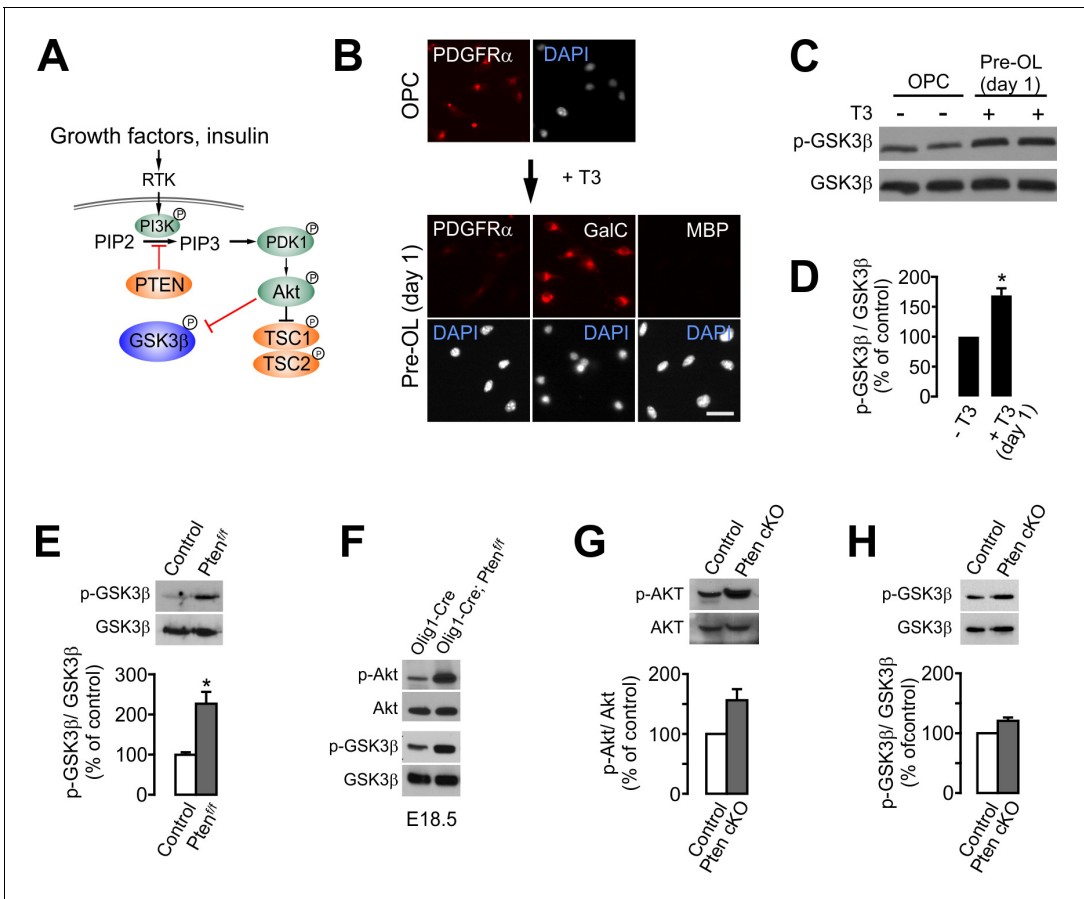

**Figure 8.** OPC-specific *Pten* ablation enhances the inhibitory phosphorylation (Ser9) of GSK3β. (**A**) Schematic diagram for the signaling flow of PTEN-Akt-GSK3β. (**B**) *In vitro* OL differentiation from OPCs with T3. One day after T3 addition, the majority of cells become pre-OLs expressing GalC, but not MBP. Scale bar, 20 µm. (**C**) Western blot analysis for phospho-GSK3β (p-GSK3β$^{Ser9}$) levels with oligodendroglial primary culture (with or without T3 incubation for 1 day). (**D**) Quantification of p-GSK3β$^{Ser9}$ levels. n = 3 independent replicates. (**E**) Western blot and quantification of p-GSK3β levels in the EYFP$^+$ cells obtained from *Pdgfra-CreER; R26-EYFP; ±Pten$^{f/f}$* mice (P20 +21) by FACS. n = 3 mice per group. (**F**) Western blot of the cortical lysates of *Olig1-Cre* (control) and *Olig1-Cre; Pten$^{f/f}$* embryo (E18.5). n = 1 mouse per group. (**G, H**) Western blot analyses and quantification of the levels of p-Akt$^{Ser473}$ (G) and p-GSK3β$^{Ser9}$ (H) in *Pten*-deleted OPCs *in vitro*. OPC primary culture was obtained from 4HT-administered *Pten$^{f/f}$* (control) or *Pdgfra-CreER; Pten$^{f/f}$* (*Pten* cKO) pups (P1 +1). n = 3 for (G, H) independent replicates. Data are represented as mean ±S.E.M. *p<0.05. Paired Student's t-test for (D, G, H). Unpaired Student's t-test for (E). The numerical data for the graphs are available in *Figure 8—source data 1*. Original western images are available in *Figure 8—source data 2*.

DOI: https://doi.org/10.7554/eLife.32021.030

The following source data is available for figure 8:

**Source data 1.** Numerical data for graphs in *Figure 8*.
DOI: https://doi.org/10.7554/eLife.32021.031

**Source data 2.** Original western blot images used for *Figure 8E–H*.
DOI: https://doi.org/10.7554/eLife.32021.032

Previously, PTEN has been only thought of as a negative regulator of OL membrane outgrowth (*Goebbels et al., 2010*; *Harrington et al., 2010*; *Maire et al., 2014*). However, neither its OPC-specific role nor its inhibitory action on the OPC-to-OL transition had been revealed. Moreover, it was thought that PTEN/Akt-initiated myelin changes are mediated mostly by mTORC1 (*Goebbels et al., 2010*; *Narayanan et al., 2009*). It is likely that the major differences between our results and those of earlier studies largely stem from the cell specificity of employed Cre drivers. Use of *Cnp* or *Plp1* promoters for Cre or CreER expression (*Goebbels et al., 2010*) may have targeted mature OL-resident *Pten*. Similarly, a constitutively active form of mutant Akt was overexpressed in OLs with *Plp1*

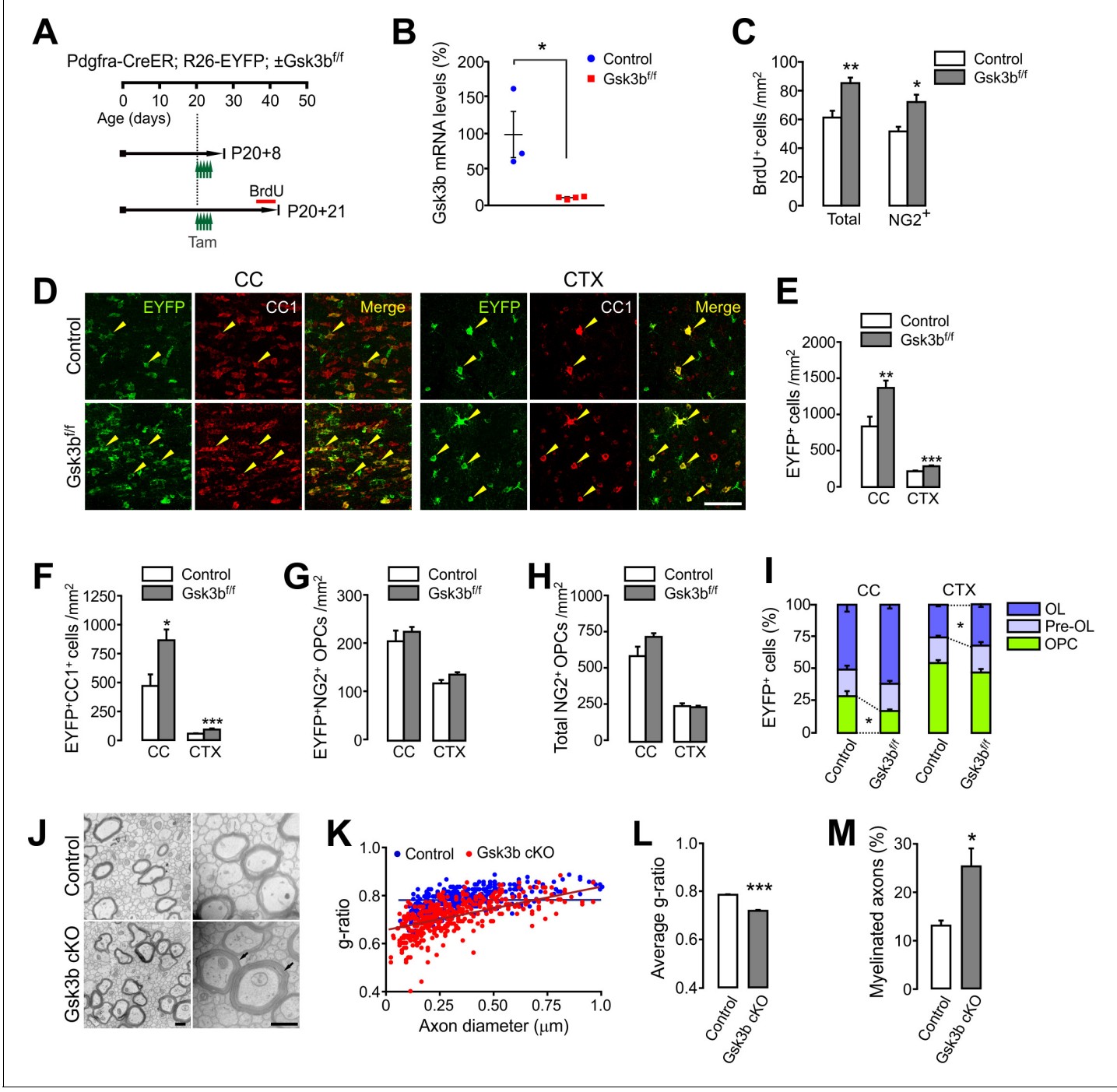

**Figure 9.** OPC-specific Gsk3b ablation enhances oligodendrocyte differentiation in the brain. (**A**) Experimental scheme for tamoxifen and BrdU administration into *Pdgfra-CreER; R26-EYFP; ±Gsk3b^f/f* and for mouse sampling. Tamoxifen (40 mg/kg per i.p. injection) was injected five times for 2.5 days starting P20. (**B**) Validation of *Gsk3b* deletion. EYFP$^+$ cells were isolated from the CTX of *Pdgfra-CreER; R26-EYFP; ±Gsk3b^f/f* mice by FACS at P20 +8. The isolated EYFP-labeled cells were subjected to RT-qPCR for *Gsk3b* mRNA levels. n = 3 (control) and 4 (*Gsk3b* cKO) mice. (**C**) Quantification of BrdU$^+$ cells in the CTX at P20 +21. There was an increases in BrdU$^+$NG2$^+$ OPCs in the *Gsk3b* cKO mice (P20 +21). n = 5 mice per group. (**D**) Confocal images of EYFP$^+$CC1$^+$ OLs (arrowheads) in the CC and CTX. Scale bar, 50 μm. (E - H) Quantification of total EYFP$^+$ cells (**E**), EYFP$^+$CC1$^+$ OLs (**F**), EYFP$^+$NG2$^+$ OPCs (**G**), and total NG2$^+$ OPCs (**H**). (**I**) Percentages of OPC, pre-OL and OL among EYFP-labeled cells. n = 8 (control) or 6 (*Gsk3b* cKO) mice for (E – I). (**J**) Representative EM of callosal axons (P13 +17). *R26-EYFP* (control) or *Pdgfra-CreER; R26-EYFP; Gsk3b^f/f* (*Gsk3b* cKO) mice were used. Scale bar, 500 nm. (**K**) Scatter plot of the g-ratios. (**L**) Average of g-ratio. At least 130 axons per mouse, three mice per group were analyzed. (**M**) Percentage of myelinated axons. Data are represented as mean ±S.E.M. n = 3 mice per group (K – M). *p<0.05; **p<0.01; ***p<0.001. Unpaired Student's t-test. The numerical data for the graphs are available in *Figure 9—source data 1*.

*Figure 9 continued*

DOI: https://doi.org/10.7554/eLife.32021.033

The following source data and figure supplements are available for figure 9:

**Source data 1.** Numerical data for graphs in *Figure 9*.

DOI: https://doi.org/10.7554/eLife.32021.037

**Figure supplement 1.** OL-specific *Gsk3b* ablation does not change the number of oligodendrocyte, but enhances expression levels of MBP and CNP.

DOI: https://doi.org/10.7554/eLife.32021.034

**Figure supplement 1—source data 1.** Numerical data for graphs in *Figure 9—figure supplement 1*.

DOI: https://doi.org/10.7554/eLife.32021.035

**Figure supplement 1—source data 2.** Original western blot images used for *Figure 9—figure supplement 1C,D*.

DOI: https://doi.org/10.7554/eLife.32021.036

promoter (*Flores et al., 2008*). Thus, those studies may not have been able to observe Akt-activated OPCs and the subsequent new OL development.

We used *Pdgfra-CreER* and Cre reporter mice to test the effects of OPC-targeted *Pten* deletion, and selectively followed the fates of affected OPCs. Although OPC-targeted gene deletion inevitably impacts OPC-derived new OLs and subsequent myelination at later time points, the phenotypic differences shown here clearly point to OPC-related mechanisms. Indeed, OL-specific *Pten* deletion with *Mog-iCre* in this study did not recapitulate OPC-specific cKO phenotypes (e.g. increases in OL number and percent of myelinated axons). Conversely, it is highly likely that the thickened myelin that was commonly observed in previous studies (*Flores et al., 2008*; *Goebbels et al., 2010*), as well as in the current study, is an OL-specific phenotype.

*Pten* has been also deleted in all Olig2-expressing cells by using *Olig2-Cre* mice, with either no change (*Harrington et al., 2010*) or a decrease (*Maire et al., 2014*) in Olig2$^+$ cells, but with thickened myelin (*Harrington et al., 2010*; *Maire et al., 2014*). However, because Olig2 is expressed in subsets of neural stem/progenitor cell populations through embryonic development, it appears that not only OL lineage cells, but also astrocytes and some types of neurons, are affected by *Olig2*-Cre (*Maire et al., 2014*; *Masahira et al., 2006*). Thus, it is possible that the phenotypes of *Olig2-Cre*; *Pten*$^{f/f}$ mice were complicated by non-cell-autonomous mechanisms (*Codeluppi et al., 2009*; *Goebbels et al., 2017*). More importantly, prolonged PTEN deficiency in OLs may have led to continuous myelin growth, which may eventually have exerted detrimental effects on the integrity of OLs and axons (*Goebbels et al., 2010*; *Harrington et al., 2010*; *Narayanan et al., 2009*), masking the positive impact of short-term PTEN inhibition on OPCs that was revealed in the present study.

PTEN is a cell-intrinsic regulator critical for axonal regeneration, and its deletion in neurons enhances post-injury axon regeneration (*Liu et al., 2010*; *Park et al., 2008*). A recent study has shown that PTEN-inactivated neurons are even able to recruit OPCs, and induce ectopic myelination in the cerebellar granule cell layer, in which myelination is normally absent (*Goebbels et al., 2017*). Our findings on PTEN's role in OPCs, along with those previous observations, have broaden our understanding of PTEN function in CNS neural repair and regeneration. One concern in targeting PTEN for better myelin repair is that long-term PTEN inhibition (or Akt over-activation) in OLs may cause axonal pathologies (*Flores et al., 2008*; *Harrington et al., 2010*). However, if PTEN inactivation were effectively guided to OPCs via short-term pharmacological delivery near the site of CNS injury, this could recruit and activate OPCs toward OL differentiation, while minimizing systemic complications. A recent *ex vivo* approach demonstrated a similar rationale by inhibiting PTEN in cultured OPCs (*De Paula et al., 2014*).

It is important to identify the downstream mechanism that mediates PTEN-inactivation-enhanced OL generation. PTEN inhibition results in phosphatidylinositol (3,4,5)-trisphosphate (PIP3) accumulation, and thus activates Akt. Even though Akt is known to phosphorylate many different downstream molecules, its major targets may differ according to the type of cell. TSC1/2 phosphorylation and subsequent mTORC1 activity has been considered the most responsible mediator of PTEN-inactivation-dependent axon regeneration (*Liu et al., 2010*; *Park et al., 2008*) and PTEN-inactivation-induced myelin membrane outgrowth (*Goebbels et al., 2010*; *Harrington et al., 2010*; *Narayanan et al., 2009*). In the PNS, PTEN also regulates the onset of myelination, exclusively through mTORC1 activity (*Figlia et al., 2017*). In most cases, the supporting evidence for the

importance of mTORC1 has been obtained with rapamycin, an mTORC1 inhibitor (*Du et al., 2015*; *Figlia et al., 2017*; *Goebbels et al., 2010*; *Narayanan et al., 2009*; *Park et al., 2008*). However, given the widespread impact of systemically administered rapamycin on multiple cell types (*Goldshmit et al., 2015*; *Srivastava et al., 2016*), the *in vivo* results may be difficult to interpret in terms of cell-type-specific mechanisms. We selectively deleted *Mtor* in OPCs, and subsequent cell fate analysis confirmed the importance of mTOR in OL development, in agreement with prior studies (*Bercury et al., 2014*; *Lebrun-Julien et al., 2014*; *Wahl et al., 2014*). It is noteworthy that the degree of impairment of OL development shown by OPC-specific *Mtor* ablation in this study was far greater than that observed with rapamycin administration (*Narayanan et al., 2009*). However, despite this effective mTOR inactivation, it did not reverse PTEN-inactivation-promoted oligodendrogenesis, indicating that the observed OL promotion was not mediated by mTOR signaling. Furthermore, when PTEN and mTOR were deleted together, the activated signaling mechanisms in OPCs sufficiently compensated for the loss of mTORC1, stimulating OL generation more than in normal OL development. Therefore, our results reveal that there are one or more PTEN-downstream mechanisms besides Akt-mTORC1 signaling that are critical for OL development and CNS myelination.

In the present study, we have shown that *Pten* ablation in OPCs also leads to the inhibitory GSK3β phosphorylation (at Ser9). Therefore, it is likely that PTEN inactivation results in GSK3β inhibition in OL lineage cells. Finally, we show that OPC-specific *Gsk3b* deletion also promotes OL development in the brain similar to that of *Pten* ablation. Interestingly, it has been shown that PTEN-inhibition (or PI3K activation) induces sensory axon regeneration, not via mTORC1 signaling (*Christie et al., 2010*; *Saijilafu et al., 2013*), but via GSK3α/β activity inhibition *in vitro* (*Saijilafu et al., 2013*). Moreover, Akt-dependent CNS axon regeneration is further promoted by GSK3β inhibition, in addition to mTORC1 signaling-dependent mechanisms (*Miao et al., 2016*), suggesting an important role of Akt-GSK3β signaling in axonal regeneration. A recent study has shown that a local delivery of GSK3β inhibitors promoted OL development from stem cells in the subventricular zone (*Azim and Butt, 2011*), although opposite *in vitro* results have also been also reported (*Zhou et al., 2014*). Therefore, an effective OPC-targeted PTEN or GSK3β inhibition may have therapeutic potential and benefit for expedited remyelination.

In summary, our study reveals that OPC-resident PTEN persistently regulates OL lineage progression, and demonstrates a beneficial potential of PTEN inhibition for better OL regeneration. Moreover, it suggests that mTORC1-dependent signaling is not the only major downstream player for these positive responses, but that PTEN-Akt-GSK3β signaling poses a negative balance, and plays an important role in OL lineage progression. These results support the idea that OPC-targeted cell-specific genetic approaches are powerful tools for identifying and evaluating cell-intrinsic molecules for the promotion of OL regeneration.

## Materials and methods

**Key resources table**

| Reagent type (species) or resource | Designation | Source or reference | Identifiers | Additional information |
|---|---|---|---|---|
| Genetic reagent (*Mus musculus*) | Pdgfra-CreER; Tg (Pdgfra-cre/ERT) | JAX #018280; PMID:21092857 | RRID:IMSR_JAX:018280 | |
| Genetic reagent (*M. musculus*) | Mog-iCre | PMID:15908920 | NA | |
| Genetic reagent (*M. musculus*) | Olig1-Cre; Olig1^wt/Cre | JAX #011105; PMID:11955448 | RRID:IMSR_JAX:011105 | |
| Genetic reagent (*M. musculus*) | R26-EYFP | JAX #006148; PMID:11299042 | RRID:IMSR_JAX:006148 | |
| Genetic reagent (*M. musculus*) | R26-mEGFP | JAX #007576; PMID:17868096 | RRID:IMSR_JAX:007576 | |
| Genetic reagent (*M. musculus*) | Mobp-EGFP; Tg (Mobp-EGFP) | MMRRC (030483-UCD); PMID:23542689 | RRID:MMRRC_030483-UCD | |

*Continued*

| Reagent type (species) or resource | Designation | Source or reference | Identifiers | Additional information |
|---|---|---|---|---|
| Genetic reagent (*M. musculus*) | Tsc1[f/f] | JAX #005680; PMID:12205640 | RRID:IMSR_JAX:005680 | |
| Genetic reagent (*M. musculus*) | Pten[f/f] | JAX #006440; PMID:11857804 | RRID:IMSR_JAX:006440 | |
| Genetic reagent (*M. musculus*) | Mtor[f/f] | JAX #011009; PMID:20008564 | RRID:IMSR_JAX:011009 | |
| Genetic reagent (*M. musculus*) | Gsk3b[f/f] | JAX #029592; PMID:18694957 | RRID:IMSR_JAX:029592 | |
| Antibody | anti-Akt (rabbit monoclonal) | Cell Signaling Technology, Danvers, MA | Cat #4691; RRID:AB_915783 | dilution (1:2,000) |
| Antibody | anti-Phospho-Akt (Ser473) (rabbit monoclonal) | Cell Signaling Technology | Cat #4060; RRID:AB_2315049 | dilution (1:2,000) |
| Antibody | anti-β-actin (mouse monoclonal) | Santa Cruz Biotechnology, Dallas, TX | Cat #sc-47778; RRID:AB_2714189 | dilution (1:2,000) |
| Antibody | anti-BrdU (rat monoclonal) | Accurate, Westbury, NY | Cat #OBT0030G; RRID:AB_609567 | dilution (1:500) |
| Antibody | anti-CC1 (APC) (mouse monoclonal) | EMD Millipore, Burlington, MA | Cat #OP80; RRID:AB_2057371 | dilution (1:70) |
| Antibody | anti-CNPase (rabbit polyclonal) | PhosphoSolutions, Aurora, CO | Cat #325-CNP; RRID:AB_2492062 | dilution (1:1,000) |
| Antibody | anti-GalC (mouse monoclonal) | EMD Millipore | Cat #MAB342; RRID:AB_2073708 | dilution (1:100) |
| Antibody | anti-GFP (goat polyclonal) | Frontier Institute, Japan | Cat #GFP-Go-Af1480; RRID:AB_2571574 | dilution (1:500) |
| Antibody | anti GFP (rabbit polyclonal) | Frontier Institute | Cat #GFP-Rb-Af2020; RRID:AB_2491093 | dilution (1:500) |
| Antibody | anti-GFP (rabbit polyclonal) | Proteintech, Chicago, IL | Cat #50430–2-AP; RRID:AB_11042881 | dilution (1:500) |
| Antibody | anti-GSK3b (rabbit monoclonal) | Cell Signaling Technology | Cat #12456 | dilution (1:2,000) |
| Antibody | anti-Phospho-GSK3β (Ser9) (rabbit monoclonal) | Cell Signaling Technology | Cat #9323; RRID:AB_2115201 | dilution (1:2,000) |
| Antibody | anti-Ki67 (rabbit polyclonal) | Abcam, Cambridge, MA | Cat #ab15580; RRID:AB_443209 | dilution (1:500) |
| Antibody | anti-MAG (mouse monoclonal) | Santa Cruz Biotechnology | Cat #sc-166849; RRID:AB_2250078 | dilution (1:2,000) |
| Antibody | anti-MBP (rabbit monoclonal) | Cell Signaling Technology | Cat #78896 | dilution (1:1,000) |
| Antibody | anti-MBP (mouse monoclonal) | Covance, Princeton, NJ | Cat #SMI-99P; RRID:AB_10120130 | dilution (1:500) |
| Antibody | anti-MOG (mouse monoclonal) | Santa Cruz Biotechnology | Cat #sc-166172; RRID:AB_2145540 | dilution (1:1,000) |
| Antibody | anti-NG2 (guinea pig polyclonal) | Gift from Dr. Dwight Bergles (Johns Hopkins) | N/A | dilution (1:4,000) |
| Antibody | anti-Olig2 (rabbit polyclonal) | EMD Millipore | Cat #AB9610; RRID:AB_570666 | dilution (1:500) |
| Antibody | anti-CD140a (PDGFRα) (Clone APA5) (rat monoclonal) | BD Biosciences, San Jose, CA | Cat #558774; RRID:AB_397117 | dilution (1:250) |
| Antibody | anti-PLP (mouse monoclonal) | EMD Millipore | Cat #MAB388-100UG; RRID:AB_177623 | dilution (1:1,000) |

*Continued on next page*

*Continued*

| Reagent type (species) or resource | Designation | Source or reference | Identifiers | Additional information |
|---|---|---|---|---|
| Antibody | Phospho-S6 ribosomal protein (Ser240/244) (rabbit polyclonal) | Cell Signaling Technology | Cat #2215; RRID:AB_331683 | dilution (1:500) |
| Antibody | Alexa488-conjugated anti-rabbit | Jackson ImmunoResearch, West Grove, PA | Cat #703-545-152 | dilution (1:500) |
| Antibody | Alexa488-conjugated anti-goat | Jackson ImmunoResearch | Cat #705-545-147 | dilution (1:500) |
| Antibody | Cy3-conjugated anti-mouse | Jackson ImmunoResearch | Cat #715-165-151 | dilution (1:500) |
| Antibody | Cy3-conjugated anti-rabbit | Jackson ImmunoResearch | Cat #711-165-152 | dilution (1:500) |
| Antibody | Cy3-conjugated anti-rat | Jackson ImmunoResearch | Cat #712-165-153 | dilution (1:500) |
| Antibody | Alexa647-conjugated anti-guinea pig | Jackson ImmunoResearch | Cat #706-605-148 | dilution (1:500) |
| Antibody | Cy5-conjugated anti-rat | Jackson ImmunoResearch | Cat #712-175-153 | dilution (1:500) |
| Antibody | HRP-conjugated anti-mouse | Jackson ImmunoResearch | Cat #715-035-150 | dilution (1:20,000) |
| Antibody | HRP-conjugated anti-rabbit | Jackson ImmunoResearch | Cat #711-035-152 | dilution (1:20,000) |
| Antibody | IR-Dye 800 CW anti-mouse | LI-COR, Lincoln, NE | Cat #926–32210 | dilution (1:5,000) |
| Antibody | Nanogold-IgG anti-rabbit | Nanoprobes, Yaphank, NY | Cat #2003 | dilution (1:100) |
| Sequence-based reagent | Gsk3b Ex1 (F) | this paper | | 5' GACCGAGAACCACCTCCTTT 3' |
| Sequence-based reagent | Gsk3b Ex2 (R) | this paper | | 5' ACTGACTTCCTGTGGCCTGT 3' |
| Sequence-based reagent | β-actin (F) | this paper | | 5' TGACAGGATGCAGAAGGAGA 3' |
| Sequence-based reagent | β-actin (R) | this paper | | 5' CGCTCAGGAGGAGCAATG 3' |
| Commercial assay or kit | TSA (+) Cyanine 3 and Fluorescein System | PerkinElmer, Hopkinton, MA | Cat #NEL753001KT | |
| Commercial assay or kit | SuperSignal West Dura | Thermo-Fisher Scientific, Waltham, MA | Cat #34075 | |
| Commercial assay or kit | Neuronal Tissue Dissociation Kit (P) | Myltenyi Biotec, Auburn, CA | Cat #130-092-628 | |
| Commercial assay or kit | RNeasy Plus Micro Kit (50) | Qiagen, Germantown, MD | Cat #74034 | |
| Commercial assay or kit | SuperScript III First-Strand synthesis system | Thermo-Fisher Scientific | Cat #18080–051 | |
| Commercial assay or kit | Pierce BCA protein assay | Thermo-Fisher Scientific | Cat #23227 | |
| Chemical compound, drug | 4-Hydroxytamoxifen | Sigma, St. Louis, MO | Cat #H-7904 | |
| Chemical compound, drug | Tamoxifen | Sigma | Cat #T-5648 | 40 mg/kg bw per injection |
| Chemical compound, drug | Odyssey Blocking Buffer (PBS) | LI-COR | Cat #927–40000 | |

*Continued on next page*

*Continued*

| Reagent type (species) or resource | Designation | Source or reference | Identifiers | Additional information |
|---|---|---|---|---|
| Chemical compound, drug | 5-Bromo-2'-deoxyuridine | Thermo-Fisher Scientific | Cat #BP-2508–5 | |
| Chemical compound, drug | QuantiTect SYBR | Qiagen | Cat #204143 | |
| Software, algorithm | GraphPad Prism 5.0 | GraphPad Software, La Jolla, CA | | |
| Software, algorithm | Image J | NIH, Bethesda, MD | | |
| Software, algorithm | StepOne software 2.1 | Thermo-Fisher Scientific | | |
| Software, algorithm | Image studio 3.1 | LI-COR | | |

## Mice

Mice homozygous for floxed alleles of *Tsc1* (*Uhlmann et al., 2002*) (STOCK Tsc1 <tm1Djk>/J, RRID: IMSR_JAX:005680), *Pten* (*Lesche et al., 2002*) (STOCK Pten <tm1Hwu>/J, RRID:IMSR_JAX:006440), *Mtor* (*Risson et al., 2009*) (STOCK Mtor <tm1.2Koz>/J, RRID:IMSR_JAX:011009), *ROSA26-EYFP* (*Srinivas et al., 2001*) (STOCK Gt(ROSA)26Sor < tm1(EYFP)Cos>/J, RRID:IMSR_JAX:006148) and *ROSA26-mEGFP* (*Muzumdar et al., 2007*) (STOCK Gt(ROSA)26Sor < tm4(ACTB-tdTomato,-EGFP) Luo>/J, RRID:IMSR_JAX:007576) were purchased from The Jackson Laboratory. *Olig1-Cre* (*Lu et al., 2002*) mice were obtained from Dr. David Rowitch (UCSF), and are now available from The Jackson Laboratory (STOCK 129S4-Olig1 < tm1(cre)Rth>/J, RRID:IMSR_JAX:011105). *Gsk3b^{f/f}* mice (*Patel et al., 2008*) were obtained from Dr. Yang Hu (Stanford University), and are now available from The Jackson Laboratory (STOCK Gsk3b < tm2Jrw>/J, RRID:IMSR_JAX:029592). *Pdgfra-CreER* mice (*Kang et al., 2010*) (STOCK Cg-Tg(Pdgfra-cre/ERT)467Dbr/J, RRID:IMSR_JAX:018280) and *Mog-iCre* mice (*Buch et al., 2005*) have been described before. The *Mobp-EGFP* mice were developed by GENSAT, and are available from MMRRC (STOCK Tg(Mobp-EGFP)IN1Gsat/Mmucd, RRID: MMRRC_030483-UCD).

Genotyping was performed using the PCR conditions recommended by The Jackson Laboratories. The genomic background of all multiple transgenic mice used here should be considered as B6SJL, C57BL/6 and 129 mixed. For all OPC fate analyses with conditional target gene deletion, age-matched relative *Pdgfra-CreER; R26-EYFP* (or *Pdgfra-CreER; R26-mEGFP*) mice were used as the control. All western blot analysis for myelin proteins, littermate homozygous mice for floxed gene (without Cre) were used as the control. For regular EM analyses, age-matched *Pten^{f/f}* or *Pten^{f/f}; Mtor^{f/f}* mice (without Cre, but some contained *R26-EYFP* assuming that it is not expressed without Cre) were used as the control. The number of the mice and genotype information are shown in individual source data files that are linked to figure legends. All animal procedures were conducted in compliance with animal protocols approved by the Institutional Animal Care and Committee (IACUC) of Temple University School of Medicine.

## Tamoxifen and BrdU administration

(Z)−4-Hydroxytamoxifen (4HT, sigma H7904) was prepared as described before (*Badea et al., 2003*; *Kang et al., 2010*). One mg of 4HT was administered per intraperitoneal (i.p.) injection, and up to two injections were given per day with at least 6 hr apart. For several sets of experiments, tamoxifen (sigma T5648) was prepared according to *Madisen et al. (2010)* and injected in a similar way to that of 4HT. Individual tamoxifen injection dose was 40 mg/kg. The total number of 4HT (or tamoxifen) injections and the first injected age for each experiment are stated in results and/or figure legends.

For cell proliferation analysis, mice received 5-bromo-2'-deoxyuridine (BrdU) injections (50 mg/kg per injection, i.p.) twice a day, in addition to BrdU-containing drinking water (1 mg/ml BrdU in tab water with 1% sucrose supplement) for the last 4 days before mouse sampling (i.e., seven injections from P38 to P41). For the experiment of long-term OPC fate analysis with control and *Pten* cKO mice (P20 +70) (*Figure 3*), the BrdU was injected between P80 and P84, (a total of 10 injections without BrdU drinking water). For lysolecithin-induced SC demyelination (*Figure 6—figure supplement 1*), BrdU was administered only via drinking water for 13 days after LPC injection.

## Lysolecithin injection

Eight days after the first tamoxifen injection was given to *Pdgfra-CreER; R26-EYFP; ±Pten^{f/f}* mice (i.e. P58), the mice were anesthetized with Ketamine (120 mg/kg) and Xylazine (8 mg/kg). After mice were fixed on a stereotaxic instrument, an L-α-Lysolecithin (L-α-Lysophosphatidylcholine or LPC in saline; Sigma L4129)-loaded Hamilton syringe was inserted into right CC (AP, +0.8 mm; ML, −0.8 mm; DV, −2.0 mm from bregma) (*Franklin and Paxinos, 2008*). The Hamilton syringe was equipped with a 33-gauge needle (45° beveled tip) attached to a motorized stereotaxic injector (Stoelting). One μl of 1% LPC was infused at a rate of 0.05 μl/min. After injection, the needle was held in place for an additional 5 min before removal.

For spinal cord demyelination, the spinal cord was exposed between lumbar spinal segment 4 and 5 (L4/L5), and stabilized with clamps (*Di Maio et al., 2011*). A 0.5 μl of 1% LPC was injected 0.5 mm deep into the dorsal column WM of the spinal cord on the right side of midline dorsal artery at a rate of 0.1 μl/min with an angle of 45 degree by using a glass micropipette and a micromanipulator and digital injector (World Precision). The needle was left in the place for additional two min to avoid backflow of injected LPC. A total of three injections were made into sites spanning the L4/L5. After recovery from anesthesia, buprenorphine (0.7 mg/kg) was administrated by a subcutaneous injection.

## Tissue processing

Mice were deeply anesthetized with sodium pentobarbital (100 mg/kg, i.p.) and briefly perfused with PBS followed by 4% paraformaldehyde (PFA in 0.1 M phosphate buffer, pH 7.4). Brains and the spinal cords were isolated and were subjected to post-fixation in 4% PFA for additional 6 hr at 4°C. The sampled tissues were incubated in 30% sucrose solution (in PBS) for at least 36 hr at 4°C for cryoprotection, and were embedded in Optimal Cutting Temperature (OCT) medium and frozen on dry ice. Thirty-five μm-thick sections were prepared using cryostat, and collected in PBS.

## Immunofluorescence

The information of antibodies and key chemical reagents used in this study is also shown in the key resources table. Tissue sections were rinsed in PBS for 5 min (twice), permeabilized in 0.3% Triton X-100 (in PBS) for 10 min, and blocked (0.3% Triton X-100, 5% normal donkey serum, 1% BSA in PBS) for 1 hr at room temperature (RT). For fixed OPC culture, coverslips were permeabilized in 0.1% Triton X-100 (in PBS) for 20 min and blocked (0.1% Triton X-100, 5% normal donkey serum). The sections were then incubated with the primary antibodies in the same blocking solution in a free-floating manner overnight at 4°C. The primary antibodies (Abs) used in the present study were rabbit anti-GFP (Frontier Institute, #GFP-Rb-Af2020, RRID:AB_2491093, 1:500 or 1:1,000 for TSA amplification), rabbit anti-GFP (Proteintech, #50430–2-AP, RRID:AB_11042881, 1:500), goat anti-GFP (Frontier Institute, #GFP-Go-Af1480, RRID:AB_2571574, 1:500), mouse anti-APC (clone CC1, EMD Millipore, #OP80, RRID:AB_2057371, 1:100), guinea pig anti-NG2 (a gift from Dr. Dwight Bergles at Johns Hopkins University, 1:4,000), mouse anti-MBP (clone SMI-99, Covance, #SMI-99P, RRID:AB_10120130, 1:500), rabbit anti-phospho-S6^{Ser240/Ser244} (Cell Signaling Technology, #2215, RRID:AB_331683, 1:500), rat anti-BrdU (Accurate, #OBT0030G, RRID:AB_609567, 1:500), and rabbit anti-Olig2 (EMD Millipore, #AB9610, RRID:AB_570666, 1:500), rabbit anti-Ki67 (Abcam, #ab15580, RRID:AB_443209, 1:500), anti-mouse GalC (Millipore, #MAB342, RRID:AB_2073708, 1:100), rat anti-PDGFRα (BD Biosciences, #558774, RRID:AB_397117, 1:250). After washing with PBS for 5 min (three times), the sections were incubated with secondary antibodies for 2 hr at RT. The secondary Abs Alexa Fluor 488-conjugated donkey anti-rabbit (1:500) or anti-goat (1:500), Cy3-conjugated donkey anti-mouse, anti-rabbit or anti-rat (1:500), Alexa-Fluor 647-conjugated donkey anti-guinea pig (1:500), and HRP-conjugated donkey anti-rabbit (1:1,000) Abs (Jackson ImmunoResearch). In all cases nuclei were co-stained with DAPI. Sections were washed in PBS for 5 min (three times) and mounted on slides with ProLong Diamond antifade mounting media (Thermo-Fisher).

For BrdU staining, prior to blocking, brain or spinal cord sections were pre-treated with 2 N HCl for 30 min at 37°C, followed by neutralization with 0.1 M sodium borate buffer (pH 8.5) for 10 min (twice). For BrdU and EYFP co-immunostaining, brain or spinal cord sections were pre-treated with 0.3% H₂O₂ for 30 min at RT before blocking. Anti-EGFP immunostaining and BrdU (plus other immunostaining) staining were carried out sequentially. Anti-EGFP staining was performed first and the

signal was further amplified with TSA plus immunofluorescence kit (PerkinElmer) according to manufacturer's instruction.

## Image acquisition and cell analysis

As the gray and white matter in the brain and spinal cord have different rates of new OL generation depending on developmental stage or examined fate mapping time window (*Kang et al., 2010*), the degree of new OL generation was analyzed for both gray and white matter areas in the CNS. Fluorescence images were captured from the motor cortex and corpus callosum from the brain, and gray and white matter from the ventral spinal cord with a Zeiss epifluorescence microscope (Axio-Imager M2) and Axiovision 7.0 or Zen software (Zeiss, Germany). Three or four sections were used per mouse, and at least three mice were used per group. Confocal images were captured with 1 μm step with a laser scanning confocal microscope (Leica TCS SP8) and LAS X software.

## Western blot analysis

Cortices were isolated from mice and snap frozen in dry ice and were stored at −80°C until use. Lysates were prepared by homogenizing the cortices in RIPA buffer (50 mM Tris pH 7.5, 150 mM NaCl, 0.5% sodium deoxycholate, 0.1% SDS, 1% NP-40) supplemented with protease inhibitor cocktail (Thermo-Fisher). Freshly sorted cells or culture were also lysed with RIPA buffer (1000 cells per μl). Protein concentration was determined using BCA protein assay kit (Thermo-Fisher). Twenty μg of protein or 10 μl of cell lysate was mixed with 2X Laemmli Sample Buffer (Bio-Rad), and were resolved with 4–15% gradient Bis-Tris gel (Bio-Rad) or 12% Bis-Tris gel by electrophoresis. Proteins were transferred to PVDF membranes (EMD Millipore). Membranes were blocked in 5% skim milk for 1 hr at RT, and were incubated with Abs for 1 hr at RT. We used mouse anti-PLP (EMD Millipore, #MAB388, RRID:AB_177623, 1:1000), rabbit anti-MBP (Cell Signaling Technology, Cat # 78896, 1:2000), rabbit anti-CNPase (Phosphosolutions, #325-CNP, RRID:AB_2492062, 1:2000), mouse anti-MAG (Santa Cruz Biotechnology, #sc-166849, RRID:AB_2250078, 1:2000), mouse anti-MOG (Santa Cruz Biotechnology, #sc-166172, RRID:AB_2145540, 1:2000), rabbit anti-GSK3β (Cell Signaling Technology, Cat # 12456, 1:2000), rabbit anti-phospho-GSK3β$^{Ser9}$ (Cell Signaling Technology, #9323, RRID:AB_2115201, 1:2,000), rabbit anti-Akt (Cell Signaling Technology, #4691, RRID:AB_915783, 1:2000), rabbit anti-phospho-Akt$^{Ser473}$ (Cell Signaling Technology, #4060, RRID:AB_2315049, 1:2000), and mouse anti-β-actin (Santa Cruz Biotechnology, #sc-47778, RRID:AB_2714189, 1:2000) Abs. After brief washing with TBS-T for 10 min (three times), the membranes were incubated with HRP-conjugated anti-rabbit or anti-mouse IgG (Jackson ImmunoResearch, 1:20,000) in 5% skim milk for 1 hr at RT. Chemiluminescent signals were detected with ECL kit (Supersignal West Dura, Thermo-Fisher). Protein bands were quantified with Image J software. In case of sequential probing with different Abs, membranes were stripped with stripping solution (2% SDS, 0.06 M Tris-HCl, pH 6.8, 0.8% 2-mercaptoethanol) for 10 min at 55°C, and washed with TBS-T three times before a subsequent probing. The full-length original western blot images are shown in source data files linked to the relevant figure legends.

For a western for PLP, protein samples were resolved with 4–15% gradient Bis-Tris gel (Bio-Rad). The PVDF membrane was blocked in Odyssey Blocking Buffer (PBS) (LI-COR) for 30 min at RT, and probed with anti-PLP and anti-β-Actin Abs overnight at 4°C. After brief washing with TBS-T, membranes were incubated with IR-Dye 800 CW goat anti-mouse (LI-COR, 1:5,000) in the same blocking buffer that was supplemented with 0.02% Tween, 0.01% SDS for 1 hr at RT. Protein bands were detected with Odyssey infrared scanner (LI-COR). Band intensity was quantified with Image studio (version 3.1) software.

## Fluorescence-activated cell sorting (FACS)

To isolate PTEN-deficient cells from the *Pten* cKO mice, we used tamoxifen-administered *Pdgfra-CreER; R26-EYFP;* ±*Pten*$^{f/f}$ (P20 +21) mice. Cerebral cortices were dissected and processed according to manufacturer's instructions (Neuronal Tissue Dissociation Kit (P), 130-092-628, Miltenyi Biotec). To remove myelin debris, the dissociated cells were resuspended in ice-cold 0.9 M sucrose solution (in HBSS, without calcium and magnesium) and centrifuged at 300 × *g* for 10 min. After myelin suspension was discarded, the pellets were resuspended in 1 ml of fresh HBSS complemented with 1 mM EDTA and 1% BSA. Cells were sorted using (BD FACSAria IIμ) at the Flow

Cytometry Core of Temple University. The sorted EYFP$^+$ cells from control and *Pten* cKO mice were used for western blot analysis. For the validation of *Gsk3b* deletion, EYFP$^+$ cells were also isolated from whole forebrains of the tamoxifen-administered *Pdgfra-CreER; R26-EYFP; ±Gsk3b$^{f/f}$* mice (P20 +8) by FACS. The *Gsk3b*-deleted cells were subjected to RNA extraction.

## Quantitative RT-PCR

Total RNA was extracted from EYFP$^+$ cells with RNeasy Plus Micro Kit (Qiagen). The first strand cDNA was synthesized using SuperScript III First-Strand Synthesis kit (Invitrogen). The RT-qPCR was performed with QuantiTect SYBR green (Qiagen) on the Applied Biosystems Step One Plus RT-PCR system. The thermal cycle profile was: 95°C for 15 min, 40 cycles of 95°C for 15 s, 55°C for 30 s and 72°C for 15 s. The primers used for *Gsk3b* mRNA detection were: Gsk3b Ex1 (F)- 5' GACCGAGAAC-CACCTCCTTT 3' and Gsk3b Ex2 (R)- 5' ACTGACTTCCTGTGGCCTGT 3'. The primers for β-Actin (*Actb*) were: b-actin (F)−5' TGACAGGATGCAGAAGGAGA 3' and b-actin (R)- 5' cgctcaggaggag-caatg 3'.

## Electron microscopy

Mice were deeply anesthetized with pentobarbital (100 m/kg, i.p) and perfused with PBS followed by 2.5% PFA, 2% glutaraldehyde (in 0.1 M phosphate buffer, pH 7.4) for regular electron microscopy (EM). After tissue isolation, samples were post-fixed for 4 hr at 4°C, and transferred to 0.1 M phosphate buffer at 4°C. The tissues were dehydrated using a graded ethanol series and embedded in EMbed-812 (EMS). Thin sections were stained with uranyl acetate and lead citrate. The obtained sections were visualized with an electron microscope (JEOL 1010 electron microscope fitted with a Hamamatsu digital camera) at 12,000X magnification. The image acquisition was performed with AMT Advantage image capture software and g ratio measurement with image J.

For immuno-EM, 4% PFA/0.1% glutaraldehyde (in 0.1 M phosphate buffer) was used for perfusion. Brains were removed from the skull and post-fixed in the same fixative for 4 hr. Microslicer sections containing the corpus callosum (60 μm in thickness) were incubated with anti-EGFP Ab (Frontier Institute) for 12 hr followed by the incubation with Nanogold-conjugated anti-rabbit IgG (Nanoprobes, Cat # 2003, 1:100) for 2 hr at RT. Immunogold was intensified with the HQ Silver Enhancement kit (Nanoprobes). The sections were post-fixed with 2% osmium tetroxide, stained with 2% uranyl acetate, dehydrated, and embedded in epoxy resin. Ultrathin sections (70 nm in thickness, Leica Ultracut) were examined using an electron microscope (H-7650, Hitachi).

## Primary OPC culture

OPC primary culture was prepared from WT mice, or *Pten* cKO and their littermate control mice according to *Chen et al. (2007)*. For *Pten* cKO mice, each P1 pup of a whole litter of *Pten$^{f/f}$;± Pdgfra-CreER* mice received a single 4HT injection (0.2 mg s.c.) at P1. At P2, genotyping was performed for the litter, and the pups were killed 24 hr after 4HT injection (i.e., P1 +1). Cortices were isolated and digested with papain and DNase I, followed by mechanical dissociation. Cells were resuspended in DMEM supplemented with 10% horse serum and 1% penicillin/streptomycin. The mixed glial cells were seeded onto T75 flask coated with poly-D-lysine, and were maintained for 7–10 days. OPCs were further enriched by differential shaking and were seeded onto poly-D-lysine-coated coverslips. Cells were kept in defined medium. To induce OPC differentiation, the culture media was supplemented with triiodothyronine (T3, 30 ng/ml). For western blot, protein lysate was prepared in RIPA buffer 24 hr later. For immunocytochemistry, the cells were fixed with 4% PFA for 15 min and rinsed three times in PBS.

## Statistical analysis

Data analysis was performed with GraphPad Prism 5.0 software. Unless specified otherwise in the figure legends, most statistical analyses were carried out using two-tailed unpaired Student's *t*-test for two group comparison, or One-way ANOVA with Bonferroni post-hoc correction for statistical evaluation of more than two groups. For the western blot analyses for GSK3β$^{Ser9}$ or Akt$^{Ser473}$ with OPC or pre-OL culture, paired student t-test was used to compare OPC and pre-OL, or control and *Pten* cKO culture lysates.

The data and error bars indicate mean ±standard error of the mean (S.E.M.), where 'n' refers to the number of mice or replicates of culture. Prior power analyses were not performed to calculate the sample size for the experiments, because the size of the effects expected was unknown before doing the experiments. However, sample sizes were estimated based on similar experiments reported in the relevant literature in the field (*Figlia et al., 2017*; *Jiang et al., 2016*). The p values, S.E.M, t values, and d.f. of each set of comparisons are shown in source data files.

## Acknowledgements

We thank Dr. Seung Baek Han for the help in spinal cord demyelination surgery, and Jason Yang and Yu Mi Jeong for their excellent technical assistance. The work was supported by grants from the Ellison Medical Foundation (AG-NS-1101–13 to SHK), the National Institutes of Health (R01NS089586 to SHK and R01NS07693 to Y-JS), Shriners Hospitals for Children (85500-PHI-14 to SHK, 84298-PHI to H-KJ and 86600 to Y-JS).

## Additional information

### Funding

| Funder | Grant reference number | Author |
|---|---|---|
| National Institute of Neurological Disorders and Stroke | R01NS089586 | Shin H Kang |
| Ellison Medical Foundation | AG-NS-1101-13 | Shin H Kang |
| Shriners Hospitals for Children | 85500-PHI-14 | Shin H Kang |
| Shriners Hospitals for Children | 84298-PHI | Hey-Kyeong Jeong |
| National Institute of Neurological Disorders and Stroke | R01NS07693 | Young-Jin Son |
| Shriners Hospitals for Children | 86600 | Young-Jin Son |

The funders had no role in study design, data collection and interpretation, or the decision to submit the work for publication.

### Author contributions

Estibaliz González-Fernández, Data curation, Formal analysis, Investigation, Writing—original draft; Hey-Kyeong Jeong, Masahiro Fukaya, Rabia R Khawaja, Isha N Srivastava, Formal analysis, Investigation; Hyukmin Kim, Investigation; Ari Waisman, Resources; Young-Jin Son, Supervision, Funding acquisition, Investigation; Shin H Kang, Conceptualization, Data curation, Supervision, Funding acquisition, Writing—original draft, Writing—review and editing

### Author ORCIDs

Young-Jin Son (iD) http://orcid.org/0000-0001-5725-9775
Shin H Kang (iD) http://orcid.org/0000-0002-3692-9802

### Ethics

Animal experimentation: All animal procedures were conducted in compliance with animal protocols (ACUP 4539 and 4568) approved by Institutional Animal Care and Committee (IACUC) at Temple University School of Medicine.

### Decision letter and Author response

Decision letter https://doi.org/10.7554/eLife.32021.039
Author response https://doi.org/10.7554/eLife.32021.040

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
