## [Decision Letter]

Thank you for submitting your article "PTEN regulates the cell lineage progression from NG2<sup>+</sup> glial progenitor to oligodendrocyte via mTOR-independent signaling" for consideration by *eLife*. Your article has been reviewed by three peer reviewers, one of whom, Beth Stevens is a member of our Board of Reviewing Editors and the evaluation has been overseen by a Senior Editor. The following individuals involved in review of your submission have agreed to reveal their identity: Teresa Wood (Reviewer #2).

The reviewers have discussed the reviews with one another and the Reviewing Editor has drafted this decision to help you prepare a revised submission.

Summary:

Gonzalez-Fernandez and colleagues examine the role of PTEN signaling in oligodendrocyte progenitor proliferation and differentiation into mature oligodendrocytes during development and early adulthood, both in the context of a healthy CNS and during remyelination. These data are important to the myelin field as recent research has focused on the role of PI3K/Akt/mTOR signaling in myelination. Their findings point to an mTOR-independent role for PTEN through GSK3β. Mice lacking PTEN specifically in OPCs exhibited continued differentiation of OPCs into mature OLs, even in early adulthood. This corresponds with increased myelin production, which is also observed in OL-specific deletion of PTEN. Interestingly, loss of mTOR did not abrogate these effects of PTEN deletion, suggesting an mTOR-independent mechanism. Additional data demonstrate instead that the effects of PTEN deletion require GSK3β.

The data are clearly presented and make important and novel contributions; however, there are several experiments which would strengthen the conclusions as summarized in essential revisions below. In addition to these revisions, there are several additional points raised by each reviewer that need to be clarified and addressed in a revised paper as summarized at the end.

Essential revisions:

1) The authors state that their Tsc1 data indicate decreased OL survival, yet do not show the data to support this statement. Is the decreased mature OL number is a result of decreased proliferation and/or differentiation? To distinguish these possibilities the authors could co-stain for NG2/caspase, CC1/caspase, and/or NG2/Ki67

2) In Figure 4, the authors examine PLP and MBP by Western blotting, and find that PLP is significantly increased but MBP is not. Given the different results, it would be more convincing of the structural integrity of increased myelin thickness if other myelin proteins were examined such as MOG, CNP, and MAG.

3) The authors show that increased mature OL numbers in PTEN cKO mice correspond to increased myelin thickness and number of myelinated axons. Is there any evidence to suggest that decreased OL numbers in TSC1 cKO mice correlate with decreased myelination, either by immuostaining or Western Blot techniques?

4) Is GSK3 responsible only for the differentiation enhancement that was observed in PTEN cKO mice, or is it also responsible for myelin production by mature OLs? Performing either immunostaining/Western blotting for myelin proteins in GSK3 cKO mice could address this. Is it possible that PTEN mediates differentiation via GSK3 but myelin production via a different mTOR independent mechanism?

5) Much of the data in the paper is based on the calculations of EYFP positive cells, which is taken as a reporter of Cre-recombinase activity. Another possibility is that knockout of the gene of interest in each could affect expression levels of EYFP protein. One can imagine a scenario where Cre-recombinase is active but EYFP is not detected. The authors could stain for Cre expression and see if they see it in cells with no EYFP.

6) Figure 1: Tsc1 deletion (which hyperactivates mTOR) in OPCs resulted in a decrease in the number of CC1+ cells, in Figure 6, the deletion of mTOR in OPCs had the same effect – less number of CC1+ cells. How do the authors interpret the result that Tsc1 deletion and mTOR deletion have the same effect? Moreover, how does the g-ratio and number of myelinated axons change with the deletion of mTOR alone?

7) The effect of GSK3β on OL development and myelination is shown nicely; however, the connection between PTEN-Akt and Gsk3β still needs to be clarified. The authors also mention this missing information in the Discussion section. In order to support this link more with the resources they have, they can check how the Gsk3β activity (phosphorylation by Akt) and the activity/level of the proteins downstream of Gsk3β change in Pten^f/f^ mouse.

8) Validation of the GSK3β^f/f^ mouse doesn't appear in the paper and should be included.

9) For the g-ratio calculations the authors should perform statistics on the biological replicates and not the individual axons counted. Can the authors please confirm that this is the case?

Reviewer #1:

Fernandez and colleagues investigate a well-known pathway that regulates myelin formation and thickness in the white matter development, mTOR. It has been known for some time that mTOR signaling regulates developmental myelination, but most experiments have examined this pathway in the context of development and not in the adult or injured brain. Here the authors performed experiments where they ablated upstream negative regulators of mTOR signaling, Tsc1 and PTEN, and examined their effects on OPC differentiation in the adult brain and following focal demyelination using lysolecithin.

The authors show that deletion of Tsc1 in OPCs leads to decreased numbers of oligodendrocyte-lineage cells (EYFP+) in the adult white matter as well as a reduction of the percentage of OLs and increase in the percentage of OPCs. Deletion of Pten lead to modest increases in oligodendrocyte lineage cell number as well as modest increases in OL generation at the expense of OPCs when induced with tamoxifen at P45. This was accompanied by increases in myelin production, but none of these effects were observed in an oligodendrocyte Pten conditional knockout. OPC ablation of Pten also leads to accelerated OL regeneration after LPC demyelination. Finally, the authors show that mTOR signaling is dispensable for these phenotypes in the Pten OPC cKO and that deletion of another downstream signaling pathway, GSK3, phenocopies the Pten OPC cKO.

These experiments show a critical divergence of Pten/mTOR signaling in OPCs versus OLs which could be important in demyelinating diseases where OPC differentiation is delayed or inefficient. This study offers a mechanistic look at the effects of different pathway components on OPC differentiation and myelination using several different conditional knockout lines. Overall the data are clearly communicated and support the conclusions. The study would be further strengthened by a few additional controls demonstrating that the cKO mice are targeting the genes as expected.

Specific comments:

The authors show validation of the Tsc1 f/f mouse using p-56 staining as a proxy for mTOR activation. These controls are performed in Figure 6 but should be moved to an earlier figure for the Pten f/f mouse. Also, validation of the GSK3b f/f mouse doesn't appear in the paper and should be included.

For the g-ratio calculations the authors should perform statistics on the biological replicates and not the individual axons counted. Can the authors please confirm that this is the case?

Reviewer #2:

Gonzalez-Fernandez and colleagues examine the role of PTEN signaling in oligodendrocyte progenitor proliferation and differentiation into mature oligodendrocytes during development and early adulthood, both in the context of a healthy CNS and during remyelination. These data are important to the myelin field as recent research has focused on the role of PI3K/Akt/mTOR signaling in myelination. The data presented here specifically point to an mTOR independent role for PTEN through GSK3β. Mice lacking PTEN specifically in OPCs exhibited continued differentiation of OPCs into mature OLs, even in early adulthood. This corresponds with increased myelin production, which is also observed in OL specific deletion of PTEN. Interestingly, loss of mTOR did not abrogate these effects of PTEN deletion, suggesting an mTOR independent mechanism. Additional data demonstrate instead that the effects of PTEN deletion require GSK3β. The data are clearly presented and make important and novel contributions. However, there are several experiments which would strengthen the conclusions.

Additionally, it is not totally clear why the TSC1 data are included, other than to suggest that TSC1 mediates effects on proliferation/differentiation via mTOR, in contrast to PTEN. However, there is a much more complete story shown for PTEN than TSC1, which fails to examine myelination, OPC proliferation, and OL survival.

1) The authors state that their Tsc1 data indicate decreased OL survival, yet do not show the data to support this statement. It is possible (as is mentioned earlier in the Results section) that the decreased mature OL number is a result of decreased proliferation and/or differentiation. Would it be possible to distinguish these possibilities by co-staining for NG2/caspase, CC1/caspase, and/or NG2/Ki67?

2) Figure 4 – The authors examine PLP and MBP by Western blotting and find that PLP is significantly increased but MBP is not. Given the different results, it would be more convincing of the structural integrity of increased myelin thickness if other myelin proteins were examined such as MOG, CNP, and MAG. Alternatively, the authors could show representative high magnification EMs showing ultrastructure.

3) The authors show that increased mature OL numbers in PTEN cKO mice correspond to increased myelin thickness and number of myelinated axons. Is there any evidence to suggest that decreased OL numbers in TSC1 cKO mice correlate with decreased myelination, either by immuostaining, Western, or EM techniques?

4) Did the mice lacking PTEN during remyelination exhibit thicker myelin or more remyelinated axons comparable to that seen during development? It is suggested in the discussion that targeting PTEN and/or GSK3B would enhance remyelination yet the increase in cell number alone does not necessarily correspond to enhanced repair.

5) With the data provided, it is difficult to assess whether GSK3B is responsible only for the differentiation enhancement that was observed in PTEN cKO mice, or if it is also responsible for myelin production by mature OLs. In the GSK3B cKO mice, would it be possible to perform either immunostaining/Western blotting for myelin proteins, or to perform EM analysis? Is it possible that PTEN mediates differentiation via GSK3B but myelin production via a different mTOR independent mechanism?

Reviewer #3:

In this manuscript Gonzalez-Fernandez and colleagues report mTOR-independent effects of PTEN loss on the development of oligodendrocytes (OLs). The role of mTOR in oligodendrocyte and Schwann cell development and myelination has been thoroughly studied in the recent years, and the research so far shows that a delicate balance of mTOR activity is required at distinct steps of OL maturation. The authors focus here on whether mTOR activity is necessary for the transition of oligodendrocyte progenitors (OPCs) to myelinating oligodendrocytes (OLs) and whether PTEN deletion has an mTOR-independent effect in this process.

The authors have done a large number of experiments using conditional and inducible knockouts of Tsc1, Pten, mTOR and GSK3b. The approach taken is elegant and the findings could be important if validated. While the quantity of the data is large, the amount of analysis of each model is not sufficiently thorough. The conclusions are based on consideration of limited possibilities and some important controls are missing. Furthermore, interpretation of the results in comparison to previously published literature makes certain assumptions that are not well supported, raising doubts about the significance of the findings.

1) A recent publication from Suter lab shows the dual function of the PI3K-Akt-mTORC1 in myelination of the PNS. They suggest a molecular mechanism where mTORC1 regulates radial sorting (high mTOR activity), and mTOR and AKT act synergistically on myelin growth and myelination. The authors should cite and discuss these results in comparison to theirs briefly in the Discussion section.

2) A previous paper reported on the loss of Tsc2 gene expression in oligodendrocytes under the Olig-2 Cre promoter (DOI:10.1002/acn3.254). The authors should include this reference as well.

3) The authors use inducible Pdgfr-CreER mice with EYFP reporter to visualize the OLs. They first knockout Tsc1 to check the effect of mTORC1 hyperactivation. They find that the number of EYFP+ cells decrease significantly in CC, but not in cortex. On the other hand, EYFP+CC1+ cells (mature OLs) decrease in both areas. In cortex then, not the OPCs but pre-OL number increases, whereas in CC both increased OPC and pre-OL numbers compensate the loss in mature OL number (Figure 1). The similar phenomenon is seen in WM of spinal cord. Can the authors comment on this distinct effect of mTOR activity on OL development in CC and cortex? Does high mTOR activity arrest OL maturation at different stages in CC and cortex?

4) Much of the data in the paper is based on the calculations of EYFP positive cells, which is taken as a reporter of Cre-recombinase activity. Another possibility is that knockout of the gene of interest in each could affect expression levels of EYFP protein. One can imagine a scenario where Cre-recombinase is active but EYFP is not detected.

5) The authors then examine whether deletion of PTEN affects the process of OL maturation and find that PTEN deletion increases the number of EYFP+ cells. In contrast to the Tsc1 deletion, they show that not in CTX but this time in CC, the number of EYFP+ OPCs do not change (Figure 2). What could be the reason for these regional differences in Tsc1 or PTEN deletion?

6) According to the results in Figure 3, they suggest that the decrease in the density of labeled OPCs in the CC may be due to rapid differentiation of PTEN-deficient OPCs into the OL, with PTEN-harboring OPCs proliferating and replacing PTEN deficient OPCs. Have they checked this by staining the overall oligos with Olig2 and looking at the changes in the number of Olig2+/Ng2+/EYFP- cells?

7) In Figure 1, they show that Tsc1 deletion (which hyperactivates mTOR) in OPCs resulted in a decrease in the number of CC1+ cells, in Figure 6, the deletion of mTOR in OPCs had the same effect – less number of CC1+ cells. How do the authors interpret the result that Tsc1 deletion and mTOR deletion have the same effect? Moreover, how does the g-ratio and number of myelinated axons change with the deletion of mTOR alone?

8) Figure 7 is missing data: the information written in the text (subsection “OPC-targeted GSK3β inactivation enhances OL development and new myelination”) is not shown in the Figure. in vitro OPC culture – stainings with Pdgfr, O4, GalC? Was this information a result of their data or does it need a citation?

9) The effect of GSK3b on OL development and myelination is shown nicely; however, the connection between PTEN-Akt and Gsk3b is still needs to be clarified. The authors also mention this missing information in the Discussion section. In order to support this link more with the resources they have, they can check how the Gsk3b activity (phosphorylation by Akt) and the activity/level of the proteins downstream of Gsk3b change in Pten(f/f) mouse.

10) A major concern is how to reconcile the data in the current manuscript with some of the previously published results. In the Discussion section, the authors argue that non-cell specific KO of PTEN does not show the same results as theirs because of non-autonomous mechanisms. There is no data to support this interpretation at this point. Importantly, if the difference is due to non-cell autonomous effect, it is going to be extremely difficult to utilize this PTEN-Akt-GSK3b axis as a means of treatment.

---

## [Author Response]

Essential revisions:1) The authors state that their Tsc1 data indicate decreased OL survival, yet do not show the data to support this statement. Is the decreased mature OL number is a result of decreased proliferation and/or differentiation? To distinguish these possibilities the authors could co-stain for NG2/caspase, CC1/caspase, and/or NG2/Ki67

As suggested, we performed co-immunostaining of EYFP/NG2/Ki67 with control and Tsc1 cKO brain sections that we had prepared.

The results indicate that OPC proliferation is not impaired in *Tsc1* cKO mice. In the CTX of *Tsc1* cKO mice, the proliferation of EYFP^+^NG2^+^ OPCs was even increased based on the % of Ki67 expression among EGFP^+^NG2^+^ OPCs. The new results were described in the text (subsection “OPC-targeted TSC1 inactivation is detrimental to new OL development”) and added to Figure 1 (as Figure 1). However, our cleaved caspase-3 staining did not clearly reveal any significant difference between the two groups, although the staining did not have any technical issues given another positive control sections (data not shown). We interpret this to mean that the P20 +21 brain sections (3 weeks after 4HT induction for cKO) may not contain many cells that are undergoing cell death. We state this in Results section.

Unfortunately, because we do not currently maintain the *Tsc1^f/f^* mouse line in our colony, we could not test some earlier time points for cell death events after 4HT induction. However, given the data showing that there are no OPC proliferation defects in the cKO, and yet no consistent OPC or pre-OL accumulation (except for CTX), our interpretation still favors impaired cell survival of mature OLs.

2) In Figure 4, the authors examine PLP and MBP by Western blotting, and find that PLP is significantly increased but MBP is not. Given the different results, it would be more convincing of the structural integrity of increased myelin thickness if other myelin proteins were examined such as MOG, CNP, and MAG.

We also performed western blot analysis again for more myelin proteins (a total of five myelin proteins) with previously prepared lysates. The new results show that levels of MOG and CNP were also increased in the CTX of OPC-specific Pten cKO mice (P16 +59). MBP increase did not reach significant levels in the new experiment. These results are added to the text (subsection “OPC-specific *Pten* ablation promotes new axon myelination”) and Figure 4.

3) The authors show that increased mature OL numbers in PTEN cKO mice correspond to increased myelin thickness and number of myelinated axons. Is there any evidence to suggest that decreased OL numbers in TSC1 cKO mice correlate with decreased myelination, either by immuostaining or Western Blot techniques?

To follow up on the reviewers’ question, we performed western blot analysis for some myelin proteins (MOG, MBP, and CNP) with spared cortical lysates (n = 3 mice per group, P20 +21). The new results show that levels of MBP were significantly decreased in *Tsc1* cKO, while the decreases of the other two protein were not significant (but there are tendencys to decrease). The new results were added to the text (subsection “OPC-targeted TSC1 inactivation is detrimental to new OL development”) and Figure 1.

4) Is GSK3B responsible only for the differentiation enhancement that was observed in PTEN cKO mice, or is it also responsible for myelin production by mature OLs? Performing either immunostaining/Western blotting for myelin proteins in GSK3B cKO mice could address this. Is it possible that PTEN mediates differentiation via GSK3B but myelin production via a different mTOR independent mechanism?

We had crossed *Gsk3b^f/f^* mice with *Mog-iCre* mice (thus, OL-specific *Gsk3b* cKO) for future analyses. In response to this question, we performed western blot of cortical lysates for myelin proteins with control (*Gsk3b^f/f^*, P31, n = 4) and cKO (*Mog-iCre;Gsk3b^f/f^* , P31, n=4). The new results indicate that levels of both MBP and CNP are increased in the cKO mice although OL-specific *Gsk3b* cKO did not change the OL numbers. These results are reminiscent of those of OL-specific *Pten* cKO mice and suggest that GSK3β signaling also inhibits not only OPC-to-OL conversion but also myelin protein synthesis. We added these data to the Result section and newly prepared Figure 9—figure supplement 1.

5) Much of the data in the paper is based on the calculations of EYFP positive cells, which is taken as a reporter of Cre-recombinase activity. Another possibility is that knockout of the gene of interest in each could affect expression levels of EYFP protein. One can imagine a scenario where Cre-recombinase is active but EYFP is not detected. The authors could stain for Cre expression and see if they see it in cells with no EYFP.

We tested one commercially available Cre Ab which did not show very strong and reliable immunoreactivities (with high background) in the brain. The reliable Cre^+^ signals were always co-localized with NG2^+^ in the CTX though. However, we have never seen any Cre signals in OLs (or in EYFP^+^CC1^+^ OLs) in the brain of 4HT-administered *Pdgfra-CreER; R26-EYFP* mice with or without *Tsc1^f/f^*(P20+21).

If we understood this point correctly, it may concern the *Tsc1* cKO phenotypes (e.g., a marked decrease in the number of EYFP^+^ OLs in corpus callosum). If so, the question would be: Is it possible that EYFP expression is suppressed (for unknown mechanisms) in OLs by TSC1 ablation? In order to test this possibility, one could check whether Cre expression remains in OLs, while EYFP expression does not. However, the Cre mice that we used in this study is *Pdgfra-CreER* mice, and this is not supposed to express Cre protein in OLs because this is OPC-specific. Unless we are testing whether these mice express Cre in off-target cells (e.g., OLs, astrocytes or neurons), Cre immunostaining or *in situ*hybridization cannot solve the question of interest.

However, if this point is concerning the issue of cell type specificity of the *Pdgfra-CreER* mice, Cre reporter-based analysis should be the best method (maybe with a different reporter than R26-EYFP, if R26-EYFP is still suspicious). Because this line of mice is for a tamoxifen-inducible Cre, even though most OPCs express the CreER, they won’t be active until tamoxifen is given. We (the corresponding author and his colleagues in his former group) have heavily tested the cell-type specificity of the *Pdgfra-CreER*Tg mouse line with different Cre reporters (Z/EG, R26-EYFP, and R26-mT/mG) and confirmed that the Cre activity is highly OPC-specific and very strong. The results were previously published (Kang et al., 2010).

6) Figure 1: Tsc1 deletion (which hyperactivates mTOR) in OPCs resulted in a decrease in the number of CC1+ cells, in Figure 6, the deletion of mTOR in OPCs had the same effect – less number of CC1+ cells. How do the authors interpret the result that Tsc1 deletion and mTOR deletion have the same effect? Moreover, how does the g-ratio and number of myelinated axons change with the deletion of mTOR alone?

In our study, both mTOR deletion and mTORC1 hyperactivation (by Tsc1 cKO) resulted in similar phenotypes regarding new OL development: OL development impairments and hypomyelination. Similar results have been reported in other studies concerning mTOR or Tsc1 cKO phenotypes (Wahl et al., 2014; Lebrun-Julien et al., 2014). Even though our study did not further investigate this issue, we agree with Lebrun-Julien et al., (2014) who suggested that overactivation of mTORC1 leads to hypomyelination in CNS, and that precisely balanced mTORC1 activity is critical for proper myelination. Alternatively, different levels of mTORC1 activities are required for different aspect of OL maturation/ myelination processes, as recently demonstrated in the case of PNS myelination (Figlia et al., 2017). However, along with the negative outcomes for myelin formation shown by other *Tsc1/ Mtor* cKO studies, our results also point to the notion that mTORC1 activity manipulation (i.e., overactivation or inhibition) may not be an effective approach for better CNS myelin repair. We add this comment in the Discussion section.

Like *Tsc1* cKO, because we do not maintain mTOR single cKO mice in our colony, we did not have available mTOR cKO mouse resource for EM. However, Wahl et al., (2014) reported relevant g-ratio data (reduced g-ratios, severe hypomyelination in the spinal cord, despite no quantification of myelinated axons) with *Cnp-Cre* based OL-specific mTOR cKO mice.

7) The effect of GSK3b on OL development and myelination is shown nicely; however, the connection between PTEN-Akt and Gsk3b still needs to be clarified. The authors also mention this missing information in the Discussion section. In order to support this link more with the resources they have, they can check how the Gsk3b activity (phosphorylation by Akt) and the activity/level of the proteins downstream of Gsk3b change in Pten(f/f) mouse.

To investigate the signaling connection between PTEN and GSK3b, we isolated PTEN-deleted EYFP^+^ cells by FACS from the control and *Pten* cKO mice (P20+21) and performed western blot for p-GSK3β. The results demonstrate that GSK3β (Ser9) phosphorylation is significantly increased by PTEN ablation in OPC or OL lineage cells (n= 3 mice per group, P< 0.05). These new results are now described in the text (subsection “PTEN ablation induces the inhibitory phosphorylation of GSK3β in OPCs”) and added to a new Figure 8 (Figure 8). Also, we showe that PTEN-deleted CTX with Olig1-Cre (so all of OPCs and some other neural cells lost PTEN) increases Akt phosphorylation and GSK3β phosphorylation in E18.5 embryos. However, this experiment was not repeated due to the difficulty in breeding and potential lethality, so statistical analysis couldn’t be applied (Figure 8).

8) Validation of the GSK3b^f/f^ mouse doesn't appear in the paper and should be included.

In response to this comment, we isolated EYFP^+^ GSK3β-deleted OPCs along with control EYFP^+^ OPCs by FACS from control (*Pdgfra-CreER; R26-EYFP*) and *Gsk3b* cKO (*Pdgfra-CreER; R26-EYFP; Gsk3b^f/f^*) mice (P20+8) and performed RT-qPCR. The results show that *Gsk3b* mRNA levels were reduced by ~ 90% in the cKO cells. This result is now depicted in new Figure 9.

9) For the g-ratio calculations the authors should perform statistics on the biological replicates and not the individual axons counted. Can the authors please confirm that this is the case?

Yes. As we made this clear in the method section and figure legends, our n number is the number of mice, not the number of axons. More than 100 axons were analyzed for the representative value of one mouse.

Reviewer #1:Fernandez and colleagues investigate a well-known pathway that regulates myelin formation and thickness in the white matter development, mTOR. It has been known for some time that mTOR signaling regulates developmental myelination, but most experiments have examined this pathway in the context of development and not in the adult or injured brain. Here the authors performed experiments where they ablated upstream negative regulators of mTOR signaling, Tsc1 and PTEN, and examined their effects on OPC differentiation in the adult brain and following focal demyelination using lysolecithin.The authors show that deletion of Tsc1 in OPCs leads to decreased numbers of oligodendrocyte-lineage cells (EYFP+) in the adult white matter as well as a reduction of the percentage of OLs and increase in the percentage of OPCs. Deletion of Pten lead to modest increases in oligodendrocyte lineage cell number as well as modest increases in OL generation at the expense of OPCs when induced with tamoxifen at P45. This was accompanied by increases in myelin production, but none of these effects were observed in an oligodendrocyte Pten conditional knockout. OPC ablation of Pten also leads to accelerated OL regeneration after LPC demyelination. Finally, the authors show that mTOR signaling is dispensable for these phenotypes in the Pten OPC cKO and that deletion of another downstream signaling pathway, GSK3, phenocopies the Pten OPC cKO.These experiments show a critical divergence of Pten/mTOR signaling in OPCs versus OLs which could be important in demyelinating diseases where OPC differentiation is delayed or inefficient. This study offers a mechanistic look at the effects of different pathway components on OPC differentiation and myelination using several different conditional knockout lines. Overall the data are clearly communicated and support the conclusions. The study would be further strengthened by a few additional controls demonstrating that the cKO mice are targeting the genes as expected.Specific comments:1) The authors show validation of the Tsc1 f/f mouse using p-56 staining as a proxy for mTOR activation. These controls are performed in Figure 6 but should be moved to an earlier figure for the Pten f/f mouse.

We thank the reviewer for this suggestion. In response to this, we added the% of p-S6+ cells among EYFP+ cells (the same parameter used in Figure 1 as the validation of Tsc1 cKO) in Pten cKO mice to Figure 2. Newly prepared confocal images and related quantification graph are depicted in Figure 2. This is the validation of PTEN inactivation and subsequent mTOR activation.

Also, validation of the GSK3b f/f mouse doesn't appear in the paper and should be included.

This point has been discussed and addressed in the essential revision section.

2) For the g-ratio calculations the authors should perform statistics on the biological replicates and not the individual axons counted. Can the authors please confirm that this is the case?

This point has been discussed and addressed in the essential revision section.

Reviewer #2:Gonzalez-Fernandez and colleagues examine the role of PTEN signaling in oligodendrocyte progenitor proliferation and differentiation into mature oligodendrocytes during development and early adulthood, both in the context of a healthy CNS and during remyelination. These data are important to the myelin field as recent research has focused on the role of PI3K/Akt/mTOR signaling in myelination. The data presented here specifically point to an mTOR independent role for PTEN through GSK3β. Mice lacking PTEN specifically in OLPs exhibited continued differentiation of OLPs into mature OLs, even in early adulthood. This corresponds with increased myelin production, which is also observed in OL specific deletion of PTEN. Interestingly, loss of mTOR did not abrogate these effects of PTEN deletion, suggesting an mTOR independent mechanism. Additional data demonstrate instead that the effects of PTEN deletion require GSK3β. The data are clearly presented and make important and novel contributions. However, there are several experiments which would strengthen the conclusions.Additionally, it is not totally clear why the TSC1 data are included, other than to suggest that TSC1 mediates effects on proliferation/differentiation via mTOR, in contrast to PTEN. However, there is a much more complete story shown for PTEN than TSC1, which fails to examine myelination, OLP proliferation, and OL survival.1) The authors state that their Tsc1 data indicate decreased OL survival, yet do not show the data to support this statement. It is possible (as is mentioned earlier in the Results section) that the decreased mature OL number is a result of decreased proliferation and/or differentiation. Would it be possible to distinguish these possibilities by co-staining for NG2/caspase, CC1/caspase, and/or NG2/Ki67?

This point has been discussed and addressed in the essential revision section.

2) Figure 4 – The authors examine PLP and MBP by Western blotting and find that PLP is significantly increased but MBP is not. Given the different results, it would be more convincing of the structural integrity of increased myelin thickness if other myelin proteins were examined such as MOG, CNP, and MAG. Alternatively, the authors could show representative high magnification EMs showing ultrastructure.

In response to the reviewer’s suggestion, we performed more western blot. This request has been addressed and discussed in the essential revision section.

3) The authors show that increased mature OL numbers in PTEN cKO mice correspond to increased myelin thickness and number of myelinated axons. Is there any evidence to suggest that decreased OL numbers in TSC1 cKO mice correlate with decreased myelination, either by immuostaining, Western, or EM techniques?

In response to the reviewer’s suggestion, we performed new western blot analysis with cortical lysates. The results and our response are described in the essential revision section.

4) Did the mice lacking PTEN during remyelination exhibit thicker myelin or more remyelinated axons comparable to that seen during development? It is suggested in the discussion that targeting PTEN and/or GSK3B would enhance remyelination yet the increase in cell number alone does not necessarily correspond to enhanced repair.

In response to this concern, EM analyses were performed, and the results suggest that g-ratios were significantly reduced for large-diameter axons (diameter > 1um), but not for smaller diameter axons. However, overall the average g-ratio was reduced in *Pten* cKO (n=3 mice), suggesting that remyelination is enhanced by *Pten* cKO. The new results are depicted in Figure 6.

5) With the data provided, it is difficult to assess whether GSK3B is responsible only for the differentiation enhancement that was observed in PTEN cKO mice, or if it is also responsible for myelin production by mature OLs. In the GSK3B cKO mice, would it be possible to perform either immunostaining/Western blotting for myelin proteins, or to perform EM analysis? Is it possible that PTEN mediates differentiation via GSK3B but myelin production via a different mTOR independent mechanism?

This is a very important question to us. We had bred mice to obtain OL-specific *Gsk3b* cKO (*Mog-iCre; Gsk3b^f/f^*) mice. Although we did not perform EM, we performed western blot analysis for myelin proteins (MAG, MBP, CNP). The levels of at least two myelin proteins (MBP and CNP) were increased in the OL-specific *Gsk3b* cKO mice. These results suggest that GSK3b also negatively regulates myelination. These new data (blot and quantification) are added to a new figure (Figure 9—figure supplement 1). This response is also described in the essential revision section (comment #4).

Reviewer #3:In this manuscript Gonzalez-Fernandez and colleagues report mTOR-independent effects of PTEN loss on the development of oligodendrocytes (OLs). The role of mTOR in oligodendrocyte and Schwann cell development and myelination has been thoroughly studied in the recent years, and the research so far shows that a delicate balance of mTOR activity is required at distinct steps of OL maturation. The authors focus here on whether mTOR activity is necessary for the transition of oligodendrocyte progenitors (OPCs) to myelinating oligodendrocytes (OLs) and whether PTEN deletion has an mTOR-independent effect in this process.The authors have done a large number of experiments using conditional and inducible knockouts of Tsc1, Pten, mTOR and GSK3b. The approach taken is elegant and the findings could be important if validated. While the quantity of the data is large, the amount of analysis of each model is not sufficiently thorough. The conclusions are based on consideration of limited possibilities and some important controls are missing. Furthermore, interpretation of the results in comparison to previously published literature makes certain assumptions that are not well supported, raising doubts about the significance of the findings.1) A recent publication from Suter lab shows the dual function of the PI3K-Akt-mTORC1 in myelination of the PNS. They suggest a molecular mechanism where mTORC1 regulates radial sorting (high mTOR activity), and mTOR and Akt act synergistically on myelin growth and myelination. The authors should cite and discuss these results in comparison to theirs briefly in the Discussion section.

We appreciate this suggestion. We cite this published work (Figlia et al., 2017) in the revision as an example showing that different mTORC1 activity levels are required for different steps of SC myelination.

2) A previous paper reported on the loss of Tsc2 gene expression in oligodendrocytes under the Olig-2 Cre promoter (DOI:10.1002/acn3.254). The authors should include this reference as well.

We cite this paper (Carson et al., 2015) in Results section of this revision.

3) The authors use inducible Pdgfr-CreER mice with EYFP reporter to visualize the OLs. They first knockout Tsc1 to check the effect of mTORC1 hyperactivation. They find that the number of EYFP+ cells decrease significantly in CC, but not in cortex. On the other hand, EYFP+CC1+ cells (mature OLs) decrease in both areas. In cortex then, not the OPCs but pre-OL number increases, whereas in CC both increased OPC and pre-OL numbers compensate the loss in mature OL number (Figure 1). The similar phenomenon is seen in WM of spinal cord. Can the authors comment on this distinct effect of mTOR activity on OL development in CC and cortex? Does high mTOR activity arrest OL maturation at different stages in CC and cortex?

We apologize if there is any confusion in the description of results. In Figure 1 (now Figure 1 in the revision), the data does not describe number of cells, but relative fraction of different stage of OL lineage cells among EYFP cells. The numbers of different OL stage cell (not OLs) were not significantly changed except for pre-OL number increase in the CTX of *Tsc1* cKO. To make it clear, we added two graphs (showing EYFP^+^ preOL numbers) to Figure 1 (for the brain) and to Figure 1—figure supplement 1 (for the spinal cord). However, EYFP^+^ OLs were so markedly and significantly decreased that the relative fractions of other stage cells (OPC and pre-OL) become larger in *Tsc1* cKO. These results suggest the two possibilities, either OL maturation arrest (between pre-OL and OL) or premature OL death. We stated these in subsection “OPC-targeted TSC1 inactivation is detrimental to new OL development”.

Based on our new analysis on NG2 OPC proliferation which was performed in response to the reviewers’ suggestion (essential revision), we exclude the possibility of OPC proliferation defects. Because there is no high level accumulation of EYFP^+^ OPCs, or EYFP^+^ pre-OLs, we interpreted that there might be premature OL death. However, our previously collected brain sections did not prove any increase in Caspase3^+^ OLs, which might be due to the too delayed observation time point. We also added this point in subsection “OPC-targeted TSC1 inactivation is detrimental to new OL development”.

4) Much of the data in the paper is based on the calculations of EYFP positive cells, which is taken as a reporter of Cre-recombinase activity. Another possibility is that knockout of the gene of interest in each could affect expression levels of EYFP protein. One can imagine a scenario where Cre-recombinase is active but EYFP is not detected.

We provide our explanation in the essential revision section.

5) The authors then examine whether deletion of PTEN affects the process of OL maturation and find that PTEN deletion increases the number of EYFP+ cells. In contrast to the Tsc1 deletion, they show that not in CTX but this time in CC, the number of EYFP+ OLPs do not change (Figure 2). What could be the reason for these regional differences in Tsc1 or PTEN deletion?

Corpus callosum (CC) is a WM area where new OL generation is very active for our fate analysis time windows. For example, between P20 and P41, oligodendrogenesis in CC is naturally very active in mice, and thus OL generation impairment (by *Tsc1* cKO) is much more easily pronounced, while enhancement in OL generation (by *Pten* cKO) is more difficult to be detected. Between P45 and P75, the OL generation usually slows down in the CC, and as a consequence, the *Pten*-cKO-induced OL promotion is clearly detected. So is in SC grey matter (P45 +30, in Figure 2—figure supplement 1).

In response to the reviewer’s point, we add statement about difference between CTX and CC in cKO phenotype penetration subsection “OPC-targeted TSC1 inactivation is detrimental to new OL development”.

6) According to the results in Figure 3, they suggest that the decrease in the density of labeled OPCs in the CC may be due to rapid differentiation of PTEN-deficient OPCs into the OL, with PTEN-harboring OPCs proliferating and replacing PTEN deficient OPCs. Have they checked this by staining the overall oligos with Olig2 and looking at the changes in the number of Olig2+/Ng2+/EYFP- cells?

In response to the reviewer’s suggestion, we also added the new Figure 3 which shows that EYFP^-^NG2^+^Olig2^+^ OPCs occupy the CC. In Figure 3, the panels F and I showed the number of EYFP+OPCs and total OPCs, respectively. Therefore, the number of EYFP^-^ OPCs could be easily calculated by subtraction (i.e., number in (I) – number in (F)). The almost all EYFP^-^ OPCs are Olig2^+^ as shown in Figure 3.

7) In Figure 1, they show that Tsc1 deletion (which hyperactivates mTOR) in OPCs resulted in a decrease in the number of CC1+ cells, in Figure 6, the deletion of mTOR in OPCs had the same effect – less number of CC1+ cells. How do the authors interpret the result that Tsc1 deletion and mTOR deletion have the same effect? Moreover, how does the g-ratio and number of myelinated axons change with the deletion of mTOR alone?

Our response to this comment is in essential revision section.

8) Figure 7 is missing data: the information written in the text (subsection “OPC-targeted GSK3β inactivation enhances OL development and new myelination”) is not shown in the Figure. in vitro OPC culture – stainings with Pdgfr, O4, GalC? Was this information a result of their data or does it need a citation?

In response to this request, we add fluorescence images showing PDGFRa, GalC, MBP expression pattern in cultured OPC (one day after T3) to new Figure 8. O4 expression is well reported in elsewhere, so we cite a reference (subsection “PTEN ablation induces the inhibitory phosphorylation of GSK3β in OPCs).

9) The effect of GSK3b on OL development and myelination is shown nicely; however, the connection between PTEN-Akt and Gsk3b is still needs to be clarified. The authors also mention this missing information in the Discussion section. In order to support this link more with the resources they have, they can check how the Gsk3b activity (phosphorylation by Akt) and the activity/level of the proteins downstream of Gsk3b change in Pten(f/f) mouse.

This is an important suggestion. For this, we performed several new western blot analysis, and provide results in new Figure 8. Our full response to this comment is shown in essential revision section.

10) A major concern is how to reconcile the data in the current manuscript with some of the previously published results. In the Discussion section, the authors argue that non-cell specific KO of PTEN does not show the same results as theirs because of non-autonomous mechanisms. There is no data to support this interpretation at this point. Importantly, if the difference is due to non-cell autonomous effect, it is going to be extremely difficult to utilize this PTEN-Akt-GSK3b axis as a means of treatment.

Utilizing Cre mice (not CreER) for loss of function studies often lead to long-term genetic inactivation through development depending on activity of the used promoter. As consequences, it is possible that there is compensation, but also it amplifies or induces secondary damages. Although we have not further sought to identify the exact causes of the phenotypic differences, our inducible Cre-based PTEN inactivation for three weeks or a month may mimic the short-term molecular inactivation, which maybe also clinically reasonable.